# In situ architecture of the human prohibitin complex

Felix Lange[1,2,7], Michael Ratz [1,6,7], Jan-Niklas Dohrke[1,2], Maxence Le Vasseur[3], Dirk Wenzel[4], Peter Ilgen [1,2,5], Dietmar Riedel [4] & Stefan Jakobs [1,2,5]✉

Prohibitins are a highly conserved family of proteins that have been implicated in a variety of functions including mitochondrial stress signalling and housekeeping, cell cycle progression, apoptosis, lifespan regulation and many others. The human prohibitins prohibitin 1 and prohibitin 2 have been proposed to act as scaffolds within the mitochondrial inner membrane, but their molecular organization has remained elusive. Here we determined the molecular organization of the human prohibitin complex within the mitochondrial inner membrane using an integrative structural biology approach combining quantitative western blotting, cryo-electron tomography, subtomogram averaging and molecular modelling. The proposed bell-shaped structure consists of 11 alternating prohibitin 1 and prohibitin 2 molecules. This study reveals an average of about 43 prohibitin complexes per crista, covering 1–3% of the crista membrane area. These findings provide a structural basis for understanding the functional contributions of prohibitins to the integrity and spatial organization of the mitochondrial inner membrane.

Prohibitin 1 (PHB1 or BAP32) and prohibitin 2 (PHB2, prohibitone, BAP37, REA) belong to the ancient and universally conserved stomatin/prohibitin/flotillin/HflK/C (SPFH) family of proteins[1,2]. A diverse set of critical functions has been attributed to the prohibitins. Although the localization of the prohibitins within the highly convoluted mitochondrial inner membrane (MIM) is undisputed, additional cellular localizations are controversially discussed[1]. On the basis of biochemical data and on negative-stain electron microscopy (EM) of prohibitin assemblies purified from yeast, it has been suggested that 12–20 PHB1/PHB2 heterodimers assemble into large ring complexes with a diameter of ~20 nm (refs. 3,4). Such ring-like structures might serve as scaffolds for proteins and lipids to form functional membrane domains. Still, direct experimental evidence for the existence, exact localization and number of ring-like prohibitin arrangements in mitochondria is missing.

Structural insights into the assembly of the SPFH family have been provided by studies expressing and purifying oligomers of the *Escherichia coli* HflK/C proteins in complex with the client bacterial AAA+ protease FtsH. Single particle cryogenic electron microscopy (cryo-EM) analysis of these reconstituted assemblies revealed that alternate HflK/C units assemble in a unique bell-shaped closed cage, containing 12 copies of each subunit[5,6]. A recent study using cryo-EM of isolated cellular vesicles demonstrated a similar cage-like assembly of flotillin-1/2, another member of the SPFH protein family, indicating that this intriguing structure resembles a common and unifying feature of the entire SPFH family[7]. However, it remained unclear whether the prohibitin subfamily adopts a different configuration.

## Results

### Prohibitins localize exclusively to mitochondria
In this study, we performed in situ cryo-EM tomography to visualize the structure of the prohibitin assembly within the MIM. Owing to the conflicting reports on the subcellular localizations of the prohibitins,

[1]Department of NanoBiophotonics, Max Planck Institute for Multidisciplinary Sciences, Göttingen, Germany. [2]Clinic of Neurology, University Medical Center Göttingen, Göttingen, Germany. [3]Altos Labs, Bay Area Institute of Science, Redwood City, CA, USA. [4]Laboratory of Electron Microscopy, Max Planck Institute for Multidisciplinary Sciences, Göttingen, Germany. [5]Fraunhofer Institute for Translational Medicine and Pharmacology ITMP, Translational Neuroinflammation and Automated Microscopy, Göttingen, Germany. [6]Present address: Department of Cell and Molecular Biology, Karolinska Institute, Stockholm, Sweden. [7]These authors contributed equally: Felix Lange, Michael Ratz. ✉e-mail: sjakobs@gwdg.de

we first aimed to characterize their localization in human U2OS cells[8,9]. We found that transient expression of PHB1 or PHB2 fused to the fluorescent protein Dreiklang (DK) resulted in aberrant mitochondrial structures or cytoplasmic localization of the fusion protein, respectively (Extended Data Fig. 1a,b)[10]. To explore whether these effects are caused by the overexpression of the prohibitins or by their tagging with a fluorescent protein, we employed clustered regularly interspaced palindromic repeats (CRISPR)–Cas9-mediated genome editing to generate human U2OS cell lines expressing PHB1 or PHB2 fused with DK from their respective native genomic loci, ensuring close to endogenous expression levels (Extended Data Figs. 2–4). The mitochondrial network of both heterozygous cell lines (PHB1–DK and PHB2–DK) displayed wild-type (WT)-like morphologies and PHB1–DK and PHB2–DK fusion proteins were localized to the mitochondria (Fig. 1a,b). This suggests that the overexpression, but not the tagging of the prohibitins with fluorescent proteins, induced artefacts.

To investigate whether endogenously expressed tagged PHB1 or PHB2 interacted with untagged prohibitins, co-immunoprecipitation experiments using nanobodies against DK were performed on mitochondria purified from knock-in cells (Extended Data Fig. 5). When PHB1–DK and PHB2–DK were used as bait proteins in co-immunoprecipitation, untagged PHB1 and PHB2, but not inner membrane proteins, such as ATP5A or COX2, were pulled down. This indicates that the tagged prohibitins formed complexes with endogenous untagged PHB1 and PHB2.

## Prohibitins are enriched at CMs

To determine the distribution of PHB1–DK and PHB2–DK within the MIM, we performed quantitative immunogold EM on both knock-in cell lines[11,12]. A highly specific antibody against DK was employed to label the target proteins. The localization of each gold particle associated with the mitochondria was assigned either to the inner boundary membrane (IBM) or the crista membrane (CM). For PHB1–DK and PHB2–DK we found 84.9% and 90.6% of the gold particles at the CMs, respectively. Additionally, with antibodies directed against endogenous PHB1 and PHB2 we found 88.6% and 88.9% of the gold particles at the CMs, respectively (Fig. 1c). When considering that the ratio of IBM to CM is 0.64:1 in the U2OS cells (Extended Data Fig. 6a), it can be calculated that the concentration of prohibitins is three to five times higher in the CMs than in the IBM (Fig. 1d). This finding was somewhat surprising as in budding yeast the prohibitins are almost evenly distributed between CM and IBM[13].

Crista-resident proteins, such as the respiratory chain complexes, have been shown to exhibit low mobility within the mitochondrial network[14–16]. To test whether PHB1 and PHB2 also exhibit low mobility, fluorescence recovery after photobleaching (FRAP) experiments were performed using genome-edited cell lines expressing either PHB1–DK or PHB2–DK (Fig. 1e,f). The fluorescence recovery was very slow for both proteins (in the minute range), with an immobile fraction ranging from 85% to 90%. In comparison, matrix-targeted DK exhibited subsecond recovery rates with an immobile fraction of <10%. These findings are consistent with the localization of the majority of prohibitins to the CMs.

## Abundance of prohibitins

Next, we aimed to determine the average number of PHB1 and PHB2 molecules per cell. To this end, recombinant His-tagged PHB1 and PHB2 proteins were purified from *E. coli* and used as a reference in quantitative western blotting (Extended Data Fig. 6b). It was determined that, on average, each U2OS cell contains approximately $(3.38 \pm 0.23) \times 10^6$ molecules of PHB1 and $(3.46 \pm 0.15) \times 10^6$ molecules of PHB2. These numbers are in good agreement with previous mass spectrometry studies using various human cell lines[17,18]. In budding yeast, prohibitins have been postulated to form ring-like complexes made of 12–20 PHB1/PHB2 dimers[3,4]. For human cells, no direct evidence for prohibitin rings was

available, but if we were to assume that human cells also contain rings of 16 prohibitin dimers, a single U2OS cell would contain approximately $2.14 \times 10^5$ prohibitin rings. As 75–84% of all PHB1/PHB2 molecules are localized to the CMs, it could be estimated that, on average, about $1.7 \times 10^5$ prohibitin rings formed by 16 dimers would be located on the CMs. To estimate how many prohibitin rings might be located on a single crista, we next determined the average number of cristae in U2OS cells. These cells contain mostly stacked lamellar cristae. To determine their average number, the average total length of the mitochondrial network and the average number of cristae per micrometre mitochondrial tubule was determined using light microscopy and EM, respectively (Extended Data Fig. 6c,d). The total mitochondrial network length of U2OS cells was $1{,}082 \pm 360$ µm and the stacked CMs had a distance of $74 \pm 15$ nm. As groups of stacked lamellar cristae are separated by membrane voids[19], which cover $21 \pm 8\%$ of the mitochondrial tubules, the corrected average distance between two CMs was $94 \pm 19$ nm (Extended Data Fig. 6e,f). Based on these measurements, an average of 11,510 cristae is estimated per U2OS cell.

On the basis of these data and the assumption that U2OS cells exhibit prohibitin rings consisting of 16 dimers, it can be calculated that each lamellar crista is expected to carry, on average, ~15 prohibitin rings, so that each crista side would exhibit 7 or 8 prohibitin rings. Assuming a ring diameter of 20 nm, these rings would cover approximately 0.7–1.4% of the available crista surface[4]. We reasoned that if in U2OS cells indeed about 1% of the cristae surface is covered by a sizeable ring-like structure, these structures should be visible on cryo-electron tomography (cryo-ET) and potentially amenable to subtomogram averaging.

Consequently, we aimed to identify about 20-nm-sized ring-shaped structures in the CMs of human cells. To this end, vitrified human U2OS cells were thinned into ~150-nm-thick lamellae using cryo-focused ion beam milling and then imaged with cryo-ET[20].

## Convex structures in the inner membrane are formed by prohibitins

We observed numerous convex structures within the MIM. The majority of these prominent structures were localized in the CMs and occasionally in the IBM (Fig. 2a(i)). From a top view, these structures appear as a ring with a diameter of about 20 nm, whereas the side view reveals a convex shape with a height of about 9 nm, with the body of the structure pointing towards the intermembrane space, that is, the crista lumen or the space between the IBM and outer membrane (Fig. 2a(ii,iii) and Extended Data Fig. 7a). We concluded that the size, localization and orientation of these structures within the inner membrane were consistent with the proposed prohibitin structure.

As prohibitins are well conserved, we suspected that similar structures should also be visible in mitochondria of other organisms. Hence, we used cryo-ET to inspect mitochondria from cultured rat hippocampal neurons and COS-7 cells derived from monkey kidney tissue. These features were also present in both cell types, indicating a common structural element within the inner membranes of mammalian mitochondria (Fig. 2b).

To investigate whether the identified structures are prohibitin complexes, we measured their abundance in tomograms taken from WT and PHB1 or PHB2 knockdown (KD) cells. Small interfering RNA-mediated silencing of either PHB1 or PHB2 downregulated the levels of both prohibitins (Extended Data Fig. 7b,c)[21,22] and strongly reduced the number of convex structures identified in PHB1 KD cells ($257 \pm 123$ particles µm$^{-3}$) and PHB2 KD cells ($202 \pm 92$ particles µm$^{-3}$) compared with WT cells ($743 \pm 328$ particles µm$^{-3}$; Fig. 2c and Extended Data Fig. 7d). We also observed one to three additional densities at most (>90%) convex structures identified in the tomograms of U2OS PHB1–DK and PHB2–DK cells (Extended Data Fig. 7e–h). These extra densities sat at the top of the convex structure but were absent in WT cells, thereby indicating that they originate from the DK fluorescent protein fused

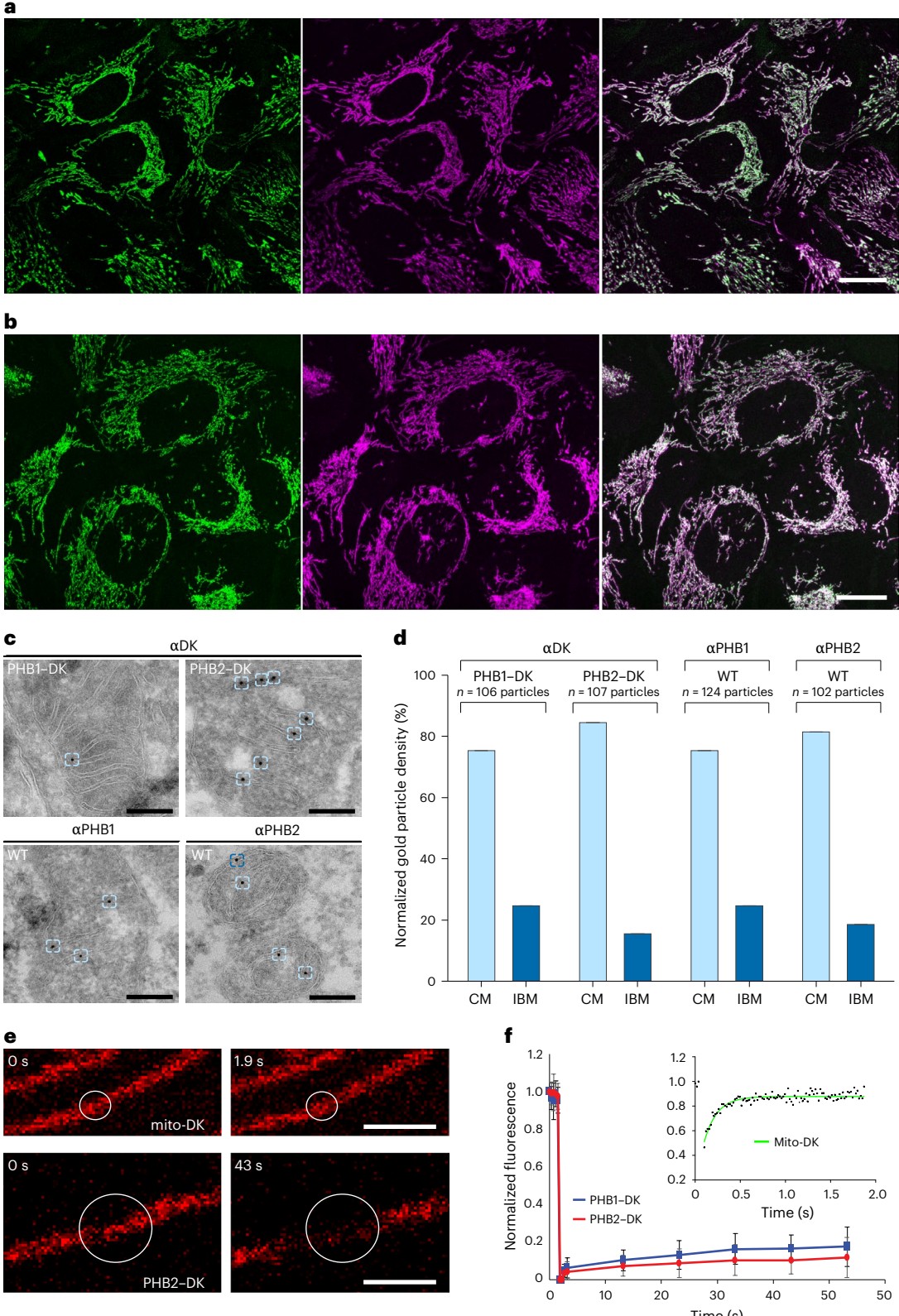

**Fig. 1 | Human PHB1 and PHB2 are primarily localized at the CMs.**
**a**,**b**, Confocal microscopy of living cells expressing PHB1–DK (**a**; $n = 3$ independent experiments) and PHB2-DK (**b**; $n = 3$ independent experiments) at endogenous levels. Mitochondria were labelled using MitoTracker deep red FM (magenta). DK fluorescence is shown in green. **c**, Immunogold EM of PHB1–DK and PHB2–DK (top) and WT (bottom) cells. Representative micrographs for each decoration with the indicated antibody are shown and the localization of

gold particles is indicated by boxes. **d**, Gold particle density in the CM and IBM (normalized to the same membrane length unit). **e**, FRAP analysis of PHB2-DK compared with mito-DK. The circle represents the area before (0 s) and after photobleaching in a representative experiment using mito-DK (1.9 s, $n = 16$ cells) or PHB2-DK (43 s, $n = 20$ cells), respectively. **f**, Recovery curves of PHB1–DK and PHB2-DK compared with mito-DK (inset). Scale bars, 10 μm (**a** and **b**), 200 nm (**c**) and 2 μm (**e**). Data are presented as mean values ± s.d.

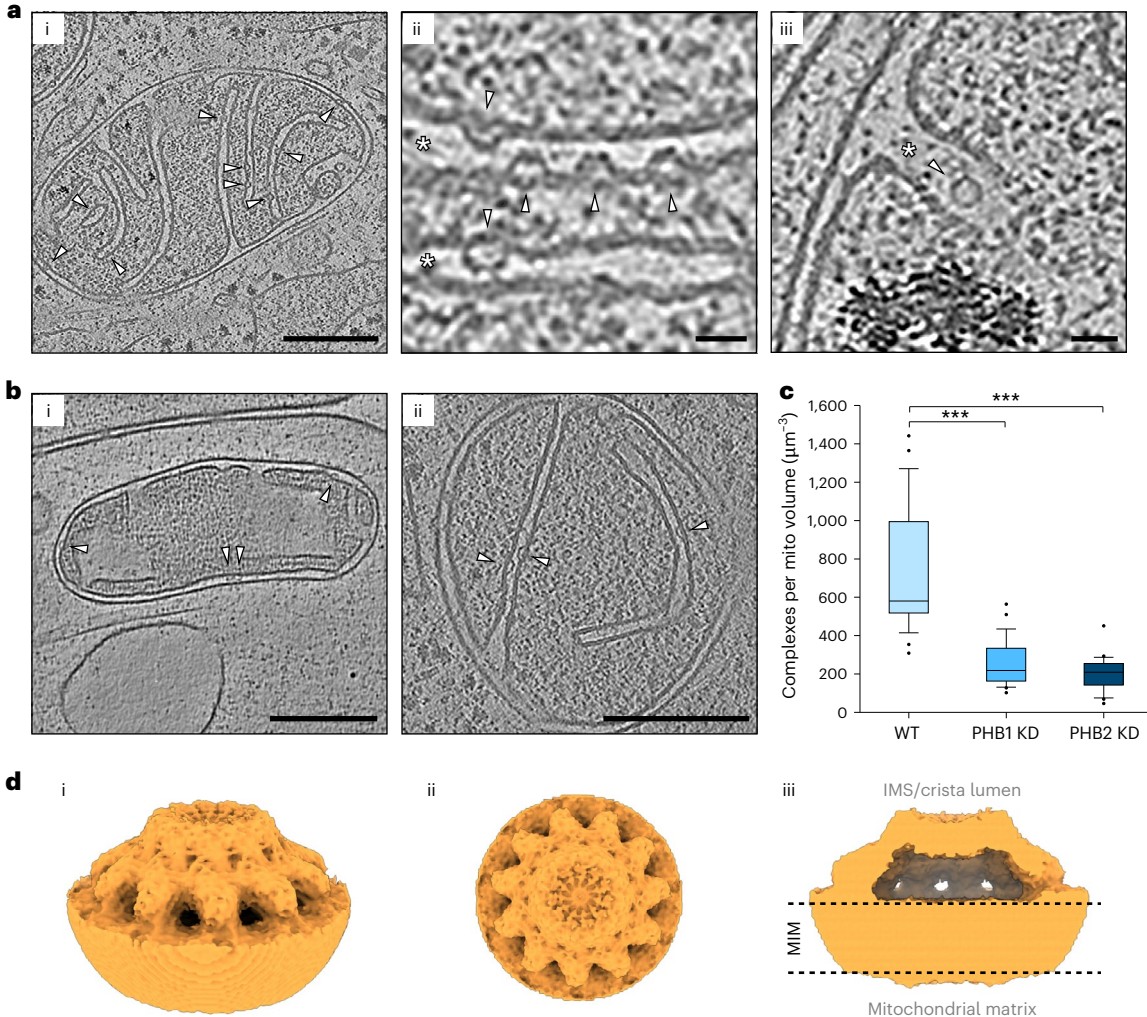

**Fig. 2 | Prohibitins form bell-shaped complexes at the MIM. a**, Cryo-tomograms of U2OS cells reveal bell-shaped assemblies at the inner membrane showing (i) a central tomographic slice of a tomogram showing multiple convex structures at the MIM (white arrowheads); (ii) a central tomographic slice at higher magnification (the convex structures feature a hollow core (side view); the lumen of the cristae is indicated with white asterisks); and (iii) a central tomographic slice at higher magnification. The structure appears as a ring (top view). **b**, Bell-shaped structures are a frequent feature of mammalian mitochondria, with (i) a central slice of an axonal mitochondrion of a rat hippocampal neuron and (ii) a central slice of a mitochondrion of a COS-7 cell. **c**, Quantification of convex complexes upon downregulation of PHB1 and PHB2. The boxes represent median, 25th and 75th percentiles, whiskers represent 10th and 90th percentiles. Three independent cultures for each condition were analysed ($n = 3$ independent experiments). Statistical comparisons using a two-sided Student's $t$-test. $P = 5 \times 10^{-7}$ and $P = 3.1 \times 10^{-8}$. **d**, Rendering of the prohibitin subtomogram average (U2OS cells, 2.5 Å pixel size) results in a round bell-shaped assembly, with (i) the side view, (ii) the top view indicating 11 units making up the bell-shaped prohibitin complex and (iii) a clipping of the side view average. The dashed grey lines indicate the location of the MIM. Scale bars, 200 nm (in **a** (i) and **b**), 25 nm (in **a** (ii) and iii)) and 10 nm (in **d**).

to the C-termini of the PHB1 and PHB2 proteins. Thus, on the basis of these observations, we concluded that the convex structures identified by cryo-ET are indeed prohibitin complexes.

**Subtomogram averaging of the prohibitin complex**

We next performed subtomogram averaging to obtain an average three-dimensional (3D) map from subvolumes[23] and gain high-resolution in situ structural information of the prohibitin complexes. To this end, we manually picked 817 particles and aligned them along the *z* axis followed by subsequent particle pose optimization in Dynamo (Extended Data Fig. 8a)[24]. An alignment of two independent half maps resulted in a resolution of 28.1 Å for the initial average (0.143 criterion, 3.97 Å pixel size; Extended Data Fig. 8b). This cryo-EM map revealed a bell-shaped structure that resembles a ring in the top view. We proceeded to extract the contrast transfer function (CTF)-corrected subtomograms in Warp for subsequent automated 3D refinement in Relion 4.0 with no initial symmetry implied[25,26]. Initially, we aligned all particles at 5 Å pixel

size. The resulting cryo-EM map revealed 11 densities in the top view, probably representing subunits of the prohibitin complex (Extended Data Fig. 8c).

On the basis of this refined cryo-EM map, we proceeded with automated 3D refinement of the dataset with C11 symmetry implied. The resulting map showed a sharper representation of the lipid bilayer and provided a clearer view on the protein bell. A Fourier shell correlation revealed a resolution of 18.3 Å for the cryo-EM map (0.143 criterion, pixel size 5 Å; Extended Data Fig. 8d,e). Finally, we re-extracted the subtomograms at 2.5 Å pixel size for automated 3D refinement with implied C11 symmetry, resulting in a final cryo-EM map of the prohibitin assembly at 16.3 Å resolution (FSC, 0.143 criterion; Fig. 2d and Extended Data Fig. 8f–h). Subsequent 3D classification in Relion 4.0 did not lead to further separation of the particles into distinct structural classes.

The final cryo-EM map reveals a circular bell-shaped structure that is embedded into the lipid bilayer with a diameter of 190.2 Å directly above the lipid bilayer. The whole bell has a height of 84.0 Å measured

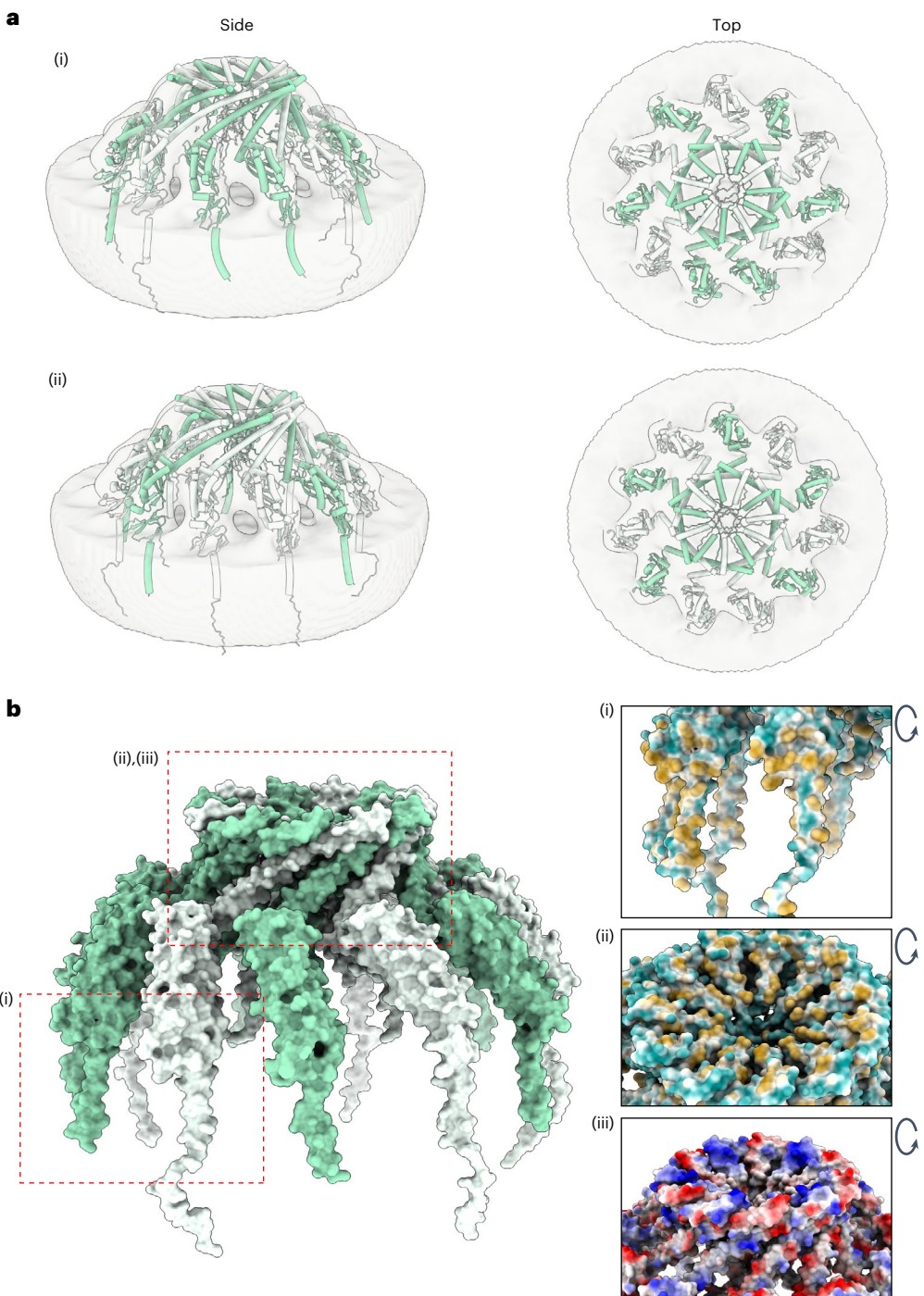

**Fig. 3 | Molecular modelling of the human prohibitin complex. a**, A cartoon representation of the predicted structures of PHB1 (green) and PHB2 (white) molecules positioned into the final cryo-EM map (light grey, transparent) in side view (left) and top view (right), with (i) a complex consisting of six PHB1 and five PHB2 molecules and (ii) a complex consisting of five PHB1 and six PHB2 molecules. **b**, The surface representation (left) and interfaces of neighbouring prohibitin molecules (right) with the hydrophobicity surface. Orange colours represent hydrophobic side residues (i and ii); the hydrophobic transmembrane domains anchor prohibitins to the membrane (i), the top of the prohibitin complex is stabilized by the hydrophobic C-termini (ii) and the electrostatic potential surface (iii); blue indicates negatively charged side chains and red indicates positively charged side chains.

from the lipid bilayer to the top of the particle and a narrowed top with a diameter of 89.0 Å. As the particles were embedded into either the IBM or the CM, the membrane opposing the top of the prohibitin bell was thus the mitochondrial outer membrane or the opposite CM of the crista. Notably, the map reveals a large empty cavity on top of the bilayer enclosed by the bell. The absence of distinct structural classes hints to no permanent occupancy of the gap by large proteins.

## The molecular architecture of the prohibitin complex
Next, we sought to model the molecular architecture of the prohibitin complex on the basis of the final cryo-EM map. As the structures of PHB1 and PHB2 have yet to be solved experimentally, we relied on AlphaFold2 to predict the structures of full length PHB1 and PHB2 (Extended Data Fig. 9a). We placed these predicted structures manually into the experimentally obtained cryo-EM map. Numerous possible

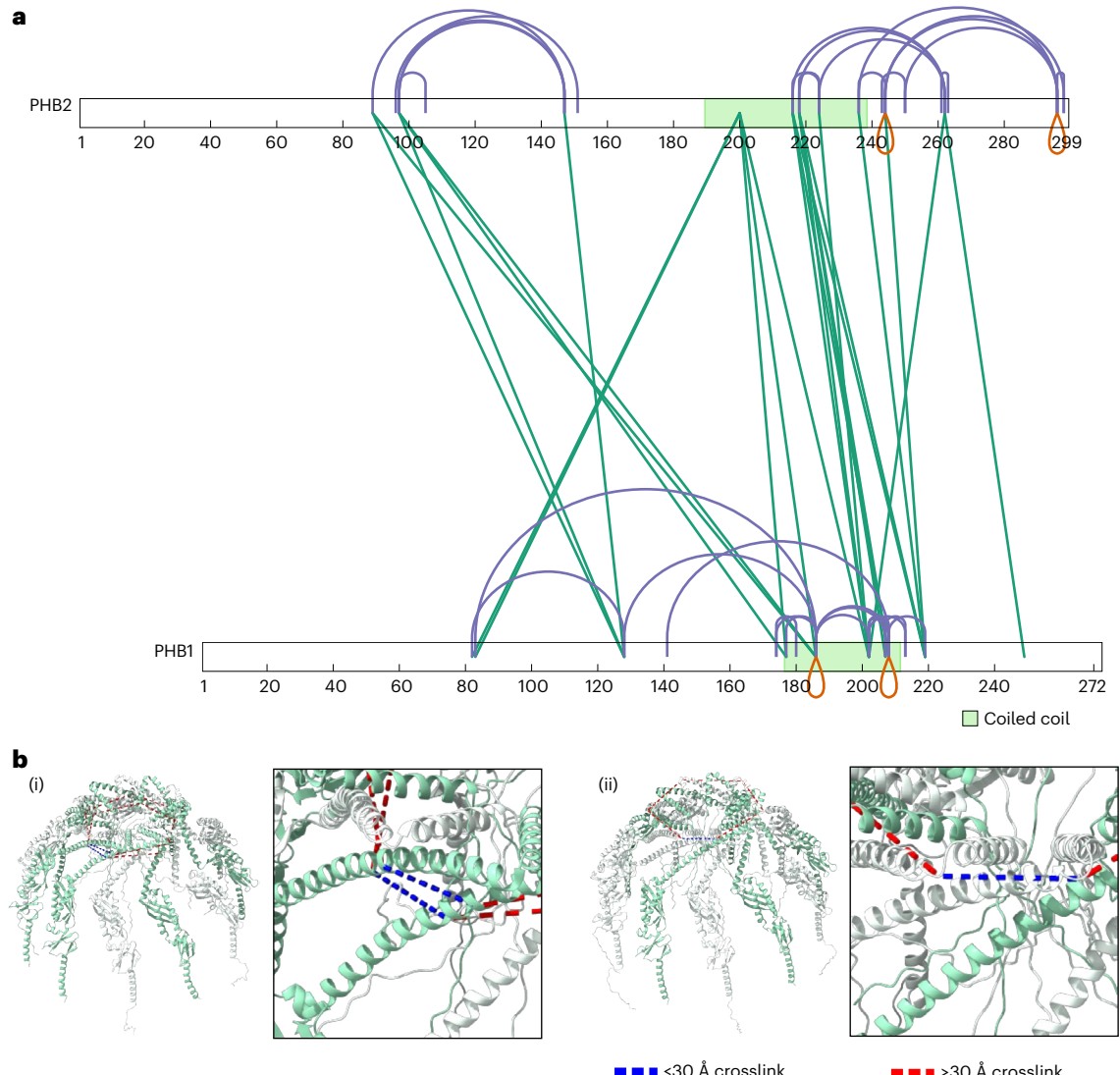

**Fig. 4 | XL-MS analysis of the prohibitin complex. a**, A schematic representation of the distribution of the intra-subunit crosslinks (magenta), inter-subunit crosslinks (green) and self-crosslinks (orange) identified in Ryl et al.[31]. The predicted coiled-coil domains are shown in green. **b**, The self-links identified were only satisfied at the PHB1–PHB1 or PHB2–PHB2 interfaces of the 6PHB1/5PHB2 (i) and 5PHB1/6PHB2 (ii) heteromeric arrangements, respectively. PHB1 is shown in green, PHB2 is shown in white, self-links <30 Å are shown in blue and unsatisfied self-links >30 Å are shown in red.

arrangements were tested but we found that each of the 11 densities in the cryo-EM map can accommodate only one prohibitin molecule. Therefore, the entire complex would consist of 11 prohibitin molecules in total. Ultimately, because biochemical evidence suggests a complex assembly with alternating PHB1 and PHB2 molecules[3], successive PHB1 and PHB2 molecules were fitted into the cryo-EM map with the N-terminal transmembrane domains embedded in the lipid bilayer[3] and the PHB domains perpendicular to the lipid bilayer. The final molecular model comprises 11 monomeric PHB molecules forming a bell-shaped assembly (Fig. 3a and Extended Data Fig. 9b,c). Obviously, the uneven symmetry implies that at least one pair of adjacent subunits comprises a repeat of the same molecule (PHB1–PHB1 or PHB2–PHB2).

According to our model, PHB1 and PHB2 are both embedded into the lipid bilayer through their N-terminal transmembrane domains in an arrangement that only supports weak interactions between neighbouring PHB N-terminal domains (Fig. 3b(i)). On the contrary, adjacent PHB C-termini interact more strongly and stabilize the centre of the bell top. The coiled-coil domains that form the bell top are stabilized by alternating negatively (PHB1: 232–252, net −3) and positively (PHB2: 244–264, net +5) charged amino acid sidechains (Fig. 3b(ii,iii) and

Extended Data Fig. 9c), consistent with previous biochemical data showing that the strongest interaction between subunits is between the helical coiled-coil domains of adjacent prohibitins[3,27].

To further test the molecular model, we queried publicly available crosslinking mass spectrometry (XL-MS) data. XL-MS provides in situ structural information by measuring the proximity of amino acids and distance restraints between protein residues[28]. By analysing crosslinked peptides from previous XL-MS studies, we identified a total of 22 and 60 unique crosslinked peptide pairs within the prohibitin heteromeric complex of mice[29,30] and humans[31], respectively. Among all crosslinks identified, 49 were unique to a single study, 12 were identified in two studies and 5 appeared in all three studies (Fig. 4a and Extended Data Fig. 10a). Owing to the remarkable degree of conservation between mouse and human PHB proteins (a single amino acid difference for PHB1 and 100% sequence identity for PHB2), all identified crosslinks could be successfully mapped to our prohibitin model. In total, 11 unique crosslinks were identified in both human and mouse species, which highlights the high structural conservation of the prohibitin complex. In good agreement with the molecular model, the majority of crosslinks mapping to the prohibitin bell top were under a 30 Å

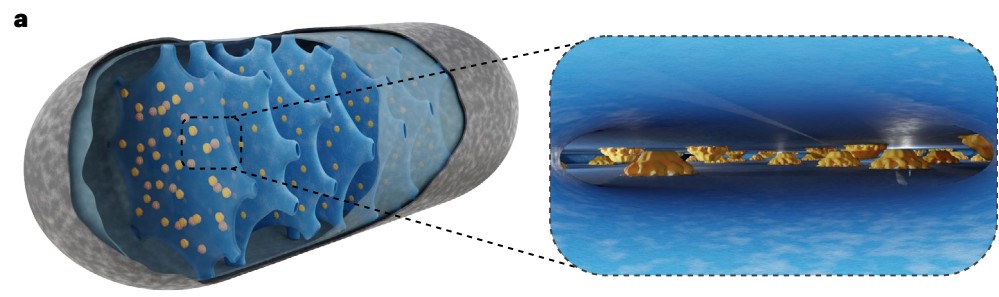

**a**

**b**

| Parameter | Average |
|---|---|
| Mitochondrial network per cell (μm) | 1,082 |
| Intercristae distance (μm) | 0.094 |
| Cristae per cell | 11,510 |
| PHB1 molecules per cell (×10⁶) | 3.38 |
| PHB2 molecules per cell (×10⁶) | 3.46 |
| PHB complexes per cell (×10⁵) | 6.22 |
| PHB complexes per crista | 43 |

**Fig. 5 | Prohibitins in the average mitochondrion. a**, A cartoon representing the abundance and distribution of prohibitin complexes in a mitochondrion. Right: the view into the crista lumen. **b**, The mitochondrial key numbers and prohibitin abundance determined in this study. The mitochondrial network length was determined in Extended Data Fig. 6c, intercristae distance determined in Extended Data Fig. 6d–f and PHB molecules per cell determined in Extended Data Fig. 6b.

$C_\alpha$–$C_\alpha$ distance, an empirically determined upper bound distance for most crosslinking studies[32] (Extended Data Fig. 10b). This reinforces the idea that the C-terminal alpha-helices of the alternating PHB proteins interact with each other to stabilize the centre of the bell top. We also observed a high number of long-distance links between the globular domains of alternating PHB1 and PHB2 proteins (Extended Data Fig. 10b). However, these crosslinks exceeded the 30 Å distance constraint, suggesting higher flexibility and large-scale protein dynamics within this region.

We also identified self-linked peptides where the same residues were crosslinked in overlapping peptides (Fig. 4b and Extended Data Fig. 10a), thereby indicating the occurrence of homomeric interactions[33]. Interestingly, these self-links only satisfied the distance constraint at the PHB1–PHB1 or PHB2–PHB2 (Fig. 4b(i,ii)) interfaces and were violated when mapped to the rest of the PHB heteromeric structure with alternating PHB1 and PHB2 subunits. These results strongly support the 11-fold symmetry model characterized by an uneven number of alternating PHB1 and PHB2 subunits. Importantly, self-links were identified in both mouse and human cells, indicating that the oligomeric organization of the prohibitin complex is conserved across species[30,31]. Overall, the crosslinking data strongly support our molecular model of the human prohibitin complex.

To test the plausibility of the final model further, we employed molecular dynamics (MD) simulations of the entire predicted complex in GROMACS (Extended Data Fig. 9d–f)[34]. The PHB domains showed fluctuations relative to the position of the globular domains. These fluctuations included small tilts of the PHB domain to the sides and are more prominent to the N-terminal part of the PHB domain, which is in line with weaker densities in the cryo-EM map at these positions (Extended Data Fig. 9g). This suggests that the N-terminal part including the PHB domain is intrinsically dynamic, while the coiled-coil domains within the prohibitin bell are more stable, consistent with our XL-MS observations. The molecular model remained stable over 100 ns of MD simulation without major alterations, suggesting that it represents a plausible model for the molecular structure of the prohibitin complex in human cells.

## Discussion

The overall bell-shaped structure of the prohibitin complex resembles the structure formed by the bacterial prohibitin homologues HflK/C[5,6]. The HflK/C structure associates with three hexameric complexes of proteases. Intriguingly, the final cryo-EM map of the prohibitin complex did not identify additional proteins, despite the fact that the complex has been demonstrated to be involved in the regulation of mitochondrial AAA proteases[35–37]. Although an association with the mitochondrial AAA protease cannot be ruled out on the basis of the subtomogram averages, the interaction with the prohibitin complex in U2OS cells may adopt a more dynamic configuration, potentially involving fewer protease molecules than the large bacterial HflK/C–FtsH complex[5,6].

As the overall bell-shaped structure formed by the prohibitins resembles the structure formed by the HflK/C and the flotillin-1/2 complexes, this arrangement seems to be a core structural principle underlying the function of SPFH proteins[5–7,38]. The molecular model of the prohibitin complex provides insight into how prohibitins establish membrane microdomains within the MIM. Quantification of the number of prohibitin complexes suggests that, on average, about 43 complexes are localized in a single crista (Fig. 5a,b). Their abundance and the fact that the top of the bell-shaped complexes is in close spatial proximity to the opposing CM or IBM might suggest an unexpected role of the prohibitin complexes in the large-scale spatial organization of CMs.

## Online content

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

## Methods

All procedures with live animals were conducted in the animal facility at the Max Planck Institute for Multidisciplinary Sciences. According to the German Animal Welfare Law, sacrificing an animal is not regarded as an experiment on animals. All requirements of section 4 TierSchG together with sections 2 Satz 2, Anlage 1, Abschnitt 2 and Anlage 2 TierSchVersV were implemented.

The facility is conducted under all aspects of animal welfare. The facility is headed by a veterinarian with special education in laboratory animal science as well as gene technology and molecular genetics.

Only professionally educated animal technicians are in charge of animal husbandry and care. The facility is registered according to section 11 Abs. 1 TierSchG (Tierschutzgesetz der Bundesrepublik Deutschland, Animal Welfare Law of the Federal Republic of Germany) as documented by 33.23-42508-066-§11, dated 16 November 2023 (*Erlaubnis, zum Halten von Wirbeltieren zur Versuchszwecken*, permission to keep vertebrates for experimental purposes) by the Niedersächsisches Landesamt für Verbraucherschutz und Lebensmittelsicherheit (Lower Saxony State Office for Consumer Protection and Food Safety). According to the Animal Welfare Law of the Federal Republic of Germany (TierSchG) and the Regulation about animals used in experiments, dated 20 December 2022 (TierSchVersV) an animal welfare officer (specialized veterinarian in laboratory animal science) and an animal welfare committee for the institute was established.

### Plasmids

**Overexpression plasmids.** Cloning of constructs for fusion protein overexpression was carried out using the primers listed in Supplementary Table 1. The DK DNA sequence was amplified via PCR from the plasmid pQE31–DK[10]. PHB1 and PHB2 coding sequences were amplified from plasmids provided by the Human ORFeome clone collection with the internal IDs 6030, 394 and 56919, respectively. The PCR products were purified and used for one-step isothermal assembly with EcoRV-digested pFLAG-CMV-5.1 (Sigma)[39].

**Nuclease plasmids.** Design of guide RNAs (gRNAs) was done using the CRISPR Design Tool (http://crispr.mit.edu). For each gRNA, a forward and reverse oligonucleotide (Supplementary Table 2) was annealed in 1x T4 ligation buffer at a final concentration of 10 µM per oligo in a thermocycler using the following parameters: 95 °C for 5 min and ramp down to 25 °C at 5 °C min$^{-1}$. The annealed oligoduplex was used in a ligation reaction for insertion into BbsI-digested pX330, a vector described elsewhere[40,41]. Plasmids containing the gRNA of interest were verified using Sanger sequencing. The final bicistronic vector encoded a gRNA and Cas9 nuclease.

**Donor plasmids.** DNA sequences for the left and right homology arms were amplified from genomic DNA using the primer pairs listed in Supplementary Table 3. The DK DNA sequence was PCR amplified from the plasmid pQE31–DK[10]. The PCR products were purified and cloned into EcoRV-digested pUC57 (Fisher Scientific) using one-step isothermal assembly followed by site-directed mutagenesis to remove Cas9 recognition sites[39].

### Cell culture

U2OS (HTB-96, American Type Culture Collection) and COS-7 cells (87021302, Sigma-Aldrich) were cultured in Dulbecco's modified Eagle's medium (DMEM) (Invitrogen) supplemented with 10% foetal bovine serum (PAA), 100 units ml$^{-1}$ penicillin, 100 µg ml$^{-1}$ streptomycin (Biochrom) and 1 mM sodium pyruvate (Sigma) under constant conditions at 37 °C and 5% $CO_2$. Plasmid transfections were done using FuGENE HD (Promega) according to the manufacturer's instructions. Mitochondria of living cells were stained using 100 nM MitoTracker Deep Red FM (Fisher Scientific) at 37 °C for 30 min.

**Cell culture for cryo-EM.** Confluent cultures of either U2OS WT or Cos7 WT cells were first detached by trypsination and subsequently suspended in DMEM complete medium. Next, 3 ml of the cell suspension was placed in ibidi glass-bottom dishes and four coated grids were added onto the bottom of the dish. Cells were allowed to adhere to the grid surface for at least 18 h before plunge freezing.

**Hippocampus isolation and dissociation.** Rat hippocampi were obtained through resection from Wistar rats. Newborn rats (P0 or P1) were decapitated and the brain was carefully extracted to preserve its integrity. The isolated brain was placed in a 3.5 cm culture dish filled with ice-cold Hank's balanced salt solution (HBSS) without calcium and magnesium (Thermo Fisher Scientific). After separating the hemispheres and removing meninges, the hippocampus was isolated, placed in a 15-ml Falcon tube and stored on ice.

For dissociation, the hippocampi were placed in 4.5 ml ice-cold HBSS without calcium and magnesium, and 0.5 ml of freshly thawed 2.5% trypsin (Thermo Fisher Scientific) was added. Dissociation occurred for 18 min in a water bath set to 37 °C with occasional gentle shaking. Digestion was halted by adding 10 ml of pre-warmed DMEM supplemented with 5% foetal bovine serum to each Falcon tube, followed by centrifugation at 100 $g$ for 3 min. After discarding the supernatant, the tissues were washed twice with 10 ml of prewarmed HBSS without calcium and magnesium.

Subsequently, tissues were dissociated using a 5 ml plastic Pasteur pipette in approximately 5 ml Neurobasal plating medium. The cell suspension was filtered through a 40 µm cell strainer into a 50 ml Falcon tube. The final cell suspension was achieved by diluting the strained cell suspension with pre-warmed Neurobasal plating medium to reach a final cell count of $1 \times 10^6$ cells ml$^{-1}$. Cells were then plated by adding 1 ml of the final cell suspension onto coated Quantifoil grids. Then, 1 h after plating, the supernatant was replaced with standard cultivation medium. Hippocampal neurons were cultivated until 18 days in vitro before freezing.

### Generation of knock-in cells

U2OS or HeLa cells were cotransfected with the respective combination of nuclease and donor plasmids for targeting either PHB1 or PHB2. At 7–10 days after transfection, the cells were subjected to single-cell sorting into 96-well plates using a FACSAria II (BD Biosciences). About 2 weeks after sorting, the cells of each well were split and equally distributed into two wells of a 24-well plate with one of the two wells containing a glass cover slip for microscopic inspection. Cells that expressed the fluorescent fusion protein were identified using an epifluorescence microscope (DM6000B, Leica Microsystems) equipped with an oil immersion objective (1.4 NA, 100×, Planapo, Leica) and a BGR filter cube (excitation: 495/15, emission: 530/30). Successfully targeted clones were expanded and genotyped via PCR using the primers listed in Supplementary Table 4.

For DNA sequencing, genomic DNA was isolated from selected clones, the respective on- or off-target site was PCR amplified (Supplementary Tables 4 and 5) followed by purification and ligation into a pCR Blunt II-TOPO vector using a Zero Blunt TOPO Kit (Thermo Fisher Scientific) according to the manufacturer's instructions. Plasmids containing an insert were identified via colony PCR and 15–20 plasmids were sequenced per locus.

### siPool-mediated KDs of prohibitins

We transfected U2OS cells with 3 nM of PHB1 or PHB2 siPools (siTOOLs Biotech), respectively following the manufacturer instructions. The cells were first cultivated for 2 days in standard six-well plates. After 2 days, the cells were detached and the cell suspension transferred to poly-L-lysine/fibronectin-coated SiO$_2$ R1/4 grids and allowed to adhere for 24 h before plunge freezing in liquid ethane at melting point. The duration of siPool KDs for both prohibitins (PHB1 and PHB2) was 3 days in total.

## Immunoblotting

Cells were grown to 80–85% confluence and detached from the growth surface and counted using a Scepter 2.0 Cell counter (EMD Millipore). Cells were collected by centrifugation followed by lysis of $10^6$ cells in 100 µl of radioimmunoprecipitation assay (RIPA) buffer supplemented with 1 mM EDTA, 1 mM PMSF, 10 U ml$^{-1}$ universal nuclease (Thermo Fisher Scientific) and 1× complete protease inhibitor cocktail (Roche). After the addition of RIPA buffer, the cell suspension was placed on ice for 30 min with vortexing steps every 10 min. The suspension was centrifuged at 16,000$g$ at 4 °C for 30 min. The supernatant was collected, and the protein concentration measured using the Pierce BCA protein assay kit (Thermo Fisher Scientific). Samples were diluted to 1.2 µg µl$^{-1}$ with RIPA buffer and mixed with the respective amount of 6× Laemmli buffer (375 mM Tris pH 6.8, 12% sodium dodecyl sulfate (SDS), 60% glycerol, 0.6 M dithiothreitol and 0.06% bromophenol blue) to a final concentration of 1 µg µl$^{-1}$. The suspension was boiled at 95 °C for 5 min, flash frozen in liquid nitrogen and stored at −20 °C for further use.

Extracts corresponding to a known number of cells as well as recombinant 6xHis-PHB1 or 6xHis-PHB2 (Abcam) were separated on 4–15% Mini-Protean TGX Precast Gels (Bio-Rad) according to the manufacturer's instructions. Separated proteins were transferred to a nitrocellulose membrane (GE Healthcare) in transfer buffer (25 mM Tris, 190 mM glycine and 20% methanol) at 4 °C for 16 h. The membrane was rinsed in Tris-buffered saline (TBS) with 0.1% Tween-20 (TBST) and incubated in 5% blocking buffer (5 g skim milk per 100 ml TBST) at room temperature (RT) for 30 min. Primary antibodies were diluted in blocking buffer and incubated with the membrane at RT for 1 h. The following primary antibodies were used: anti-PHB1 (EP2803Y, 1:2,000, Abcam), anti-PHB2 (EPR14523, 1:5,000, Abcam), anti-GFP (JL-8, 1:3,000, Clontech), anti-Actin (AC74, 1:3,000, Sigma-Aldrich) anti-COX2 (ab203912, 1:500, Abcam), anti-ATP5A (ab14748, 1:1,000, Abcam), anti-tubulin (ab15246, 1:1,000, Abcam) and anti-ESR1 (sc-8005, 1:1000, Santa Cruz Biotechnology). After washing with TBST, the membranes were incubated at RT for 1 h with horseradish peroxidase-conjugated anti-rabbit or anti-mouse secondary antibodies (Dianova) diluted 1:5,000 in blocking buffer. After washing with TBST, the membrane was incubated with Pierce ECL western blotting substrate (Thermo Fisher Scientific) and exposed to a charge-coupled device (CCD) camera. Membranes were stripped by incubation with Restore (Thermo Fisher Scientific, Waltham) at 37 °C for 30 min followed by the described protocol for reprobing with a different antibody.

## Isolation of mitochondria and immunoprecipitation

Mitochondria were isolated from WT, PHB1–DK or PHB2–DK U2OS cells grown to 80–85% confluence, detached from the growth surface and collected by centrifugation. The cell pellet was resuspended in trehalose/hepes/EDTA (THE) buffer (300 mM trehalose, 10 mM 4-(2-hydroxyethyl)-1-piperazineethanesulfonic acid–KOH, pH 7.7, 10 mM KCl and 1 mM EDTA) supplemented with 0.1% (w/v) bovine serum albumin (BSA) and lysed using a Dounce homogenizer. Homogenized cells were subjected to centrifugation at 11,000$g$ at 4 °C for 10 min. The mitochondria-containing pellet was resuspended in THE buffer without BSA and the protein concentration was measured using Bradford Protein Assay (Bio-Rad). The protein concentration was adjusted to 10 µg µl$^{-1}$ with THE buffer.

GFP-Trap-Agarose beads (Chromo-Tek) were equilibrated in ice-cold dilution buffer (20 mM Tris–HCl pH 7.5, 50 mM NaCl, 0.5 mM EDTA, 10% (v/v) glycerol, 0.3% (w/v) digitonin and 1 mM PMSF) and mixed with 800 µg of mitochondrial extract. After rotation at 4 °C for 1 h, beads were centrifuged and washed ten times using dilution buffer. The beads were resuspended in 2× Laemmli buffer (125 mM Tris pH 6.8, 4% (w/v) SDS, 20% (v/v) glycerol, 0.2 M dithiothreitol and 0.02% (w/v) bromophenol blue), boiled at 95 °C and centrifuged. The supernatant was used for subsequent SDS–polyacrylamide gel electrophoresis and western blot analysis.

## Fluorescence microscopy

**Sample preparation.** Cells were cultured on glass cover slips until they reached a confluence of about 70–85% and fixed in 37 °C prewarmed 4% (w/v) formaldehyde in PBS at RT for 5 min. The cells were permeabilized using 0.5% (v/v) Triton-X-100 in PBS for 5 min followed by subsequent incubation in blocking buffer (5% (w/v) BSA in PBS containing 100 mM glycin) for 15 min. Primary antibodies were diluted in blocking buffer and cover slips were incubated with that solution at RT for 1 h. The following primary antibodies were used: rabbit anti-PHB1 (EP2803Y, 1:250, Abcam), rabbit anti-PHB2 (EPR14523, 1:500, Abcam), mouse anti-ESR1 (D12, 1:500, Santa Cruz Biotechnology) and rabbit anti-GFP (ab290, 1:1,000, Abcam). After three washing steps in PBS, fluorophore-coupled secondary antibodies were diluted 1:1,000 and added for incubation at RT for 1 h. The following secondary antibodies were used: sheep anti-mouse and goat anti-rabbit (Dianova) coupled to KK114 (ref. [42]) or Alexa 594 (Atto-Tec). After three PBS washing steps, cells were embedded in Mowiol 4-88 mounting medium containing 1 µg ml$^{-1}$ 4,6-diamidino-2-phenylindole and 2.5% (w/v) 1,4-diazabicyclo-[2,2,2]-octane.

**Confocal microscopy.** Confocal imaging was done using the Leica TCS SP8 Confocal Microscope (Leica). All recordings were performed using a pinhole diameter of one Airy unit (1.22λ/NA), a scan speed of 400 Hz and a 63× oil immersion objective (HCX PL APO CS 63×/1.40-0.60 oil). The following laser lines were used for fluorescence excitation: a 405 Diode (405 nm), an argon laser (458 nm/476 nm/488 nm/496 nm/514 nm) and a helium–neon laser (633 nm). Fluorescence detection was done using photomultipliers and a hybrid detector operated within the dynamic range. Separation of excitation and emission light was accomplished using an acousto-optic tunable filter. Multicolour imaging was done using sequential acquisition between frames. For image digitization a sampling rate according to the Nyquist criterion was chosen. Each image was recorded at least twice for averaging.

**FRAP analysis.** FRAP measurements were done using a Leica TCS SP5 Confocal Microscope and the FRAP Wizard application (Leica). All recordings were performed using an open pinhole and a 63× oil immersion objective (HCX PL APO CS 63×/1.40-0.60 oil). Living U2OS cells were mounted in a custom-built live-cell chamber and maintained at 37 °C in $CO_2$-independent Leibovitz L-15 medium (Thermo Fisher Scientific) for imaging.

Mitochondria of U2OS cells expressing mitochondrial matrix targeted fluorescent protein Dreiklang (mito-DK) were photobleached in a circular region of interest with a diameter of about 0.5 µm using an argon laser at 20% laser power. Imaging after bleaching was performed at a time interval of 19 ms for about 2 s with the 514-nm line of the argon laser. Mitochondria of U2OS knock-in cells expressing PHB1–DK or PHB2–DK were bleached in a region of interest with a diameter of about 2.5 µm using an argon laser at 20% laser power. Imaging after bleaching was performed at a time interval of 8 s for about 1 min with the 514 nm line of the argon laser.

## EM

**Plastic embedding.** U2OS cells were grown on Aclar polymer cover slips until 80–85% confluence. Cells were prefixed in 2.5% (w/v) glutaraldehyde in 0.1 M sodium cacodylate (pH 7.4) at RT for 15 min postfixed in the same buffer at 4 °C for 15 h. Cells were washed three times in 0.1 M sodium cacodylate (pH 7.4) and incubated in 1% (w/v) $OsO_4$ in 0.1 M sodium cacodylate (pH 7.4) for 3 h. Cells were washed once in 0.1 M sodium cacodylate (pH 7.4) and then twice in water. The cells were place in 0.1% (w/v) uranyl acetate (in $H_2O$) for 30 min. Uranyl acetate was washed out by subjecting the cells to 30% ethanol three times for 5 min followed by dehydration through a 50%, 70% and 100% ethanol series. Afterwards, the cells were placed in 100% propylene oxide for 5 min and then transferred to 50%/50% propylene oxide/Epon for

1 h followed by placement to 100% Epon overnight. Samples were sectioned to 50 nm thickness with a Leica EM UC6 ultramicrotome (Leica EM UC6, Leica Microsystems). Each section was transferred to 0.7% (w/v) Pioloform-coated 200 mesh carbon grids. Samples were subjected to post contrasting using 1% (w/v) uranyl and lead acetate. EM recordings were acquired using a Philips CM 120 transmission electron microscope equipped with a TVIPS 2K × 2K slow-scan CCD camera (Philips).

**Immunogold labelling.** U2OS cells were grown to 80–85% confluence and fixed in 37 °C prewarmed 4% (w/v) formaldehyde in PBS at RT for 30 min. Further sample processing was done according to previous work[12]. Samples were sectioned into 80-nm-thin slices and incubated with diluted primary antibodies for 30 min. The following antibodies were used: anti-GFP (JL-8; 1:20, Clontech), anti-PHB1 (EP2803Y, 1:20, Abcam) and anti-PHB2 (EPR14523, 1:40, Abcam). Subsequently each sample was incubated with protein A coupled to 10 nm gold particles for 20 min followed by multiple washing steps and additional contrasting using uranyl acetate/methylcellulose on ice for 10 min. EM recordings were done using a Philips CM 120 transmission electron microscope equipped with a TVIPS 2K × 2K slow-scan CCD camera (Philips).

#### Determination of crista area occupied by PHB1/2 complexes

First, we determined that the inner diameter of an average mitochondrial tubule in human U2OS cells is 564 nm (Extended Data Fig. 6g). Thus, the area of a circular CM ($A = \pi r^2$) is around 249,832 nm². Each lamellar crista consists of two opposing membranes hence the total membrane area per crista is around 499,664 nm². Second, we assumed a ring-like prohibitin complex with a diameter of about 20 nm which corresponds to an area of about 314 nm² per prohibitin ring[2,4]. Thus, a single prohibitin ring occupies 0.063% of the total available crista area while 11–22 prohibitin rings per crista occupy between 0.7% and 1.4% of the available crista surface, respectively.

#### Cryo-EM

**Grid preparation for cell culture.** Quantifoil R2/1 or SiO₂ R1/4 gold grids (Quantifoil Mirco Tools GmbH) were briefly washed in chloroform and placed on drops of poly-D-lysine (100 µg ml⁻¹). After incubation for 1 h at RT the grids were washed three times on drops of HBSS for 10 min each. Subsequently, the grids were placed on drops of fibronectin (20 µg ml⁻¹) and incubated for 1 h at RT. Finally, the grids were washed three times on drops of HBSS for 10 min each and used directly for cell plating.

**Preparation of frozen-hydrated cells.** Grids containing cells were picked up with a pair of tweezers and any residual medium was removed by manual blotting at the tweezer tips. The sample was then rapidly loaded into a Vitrobot (Thermo Fisher) set at 37 °C and 95% humidity. We set a wait time of 120 s and applied 3 µl of a solution containing 10% glycerol in HBSS (Thermo Fisher). The grids were then backside blotted for 30 s with blot force 20, and subsequently plunged into liquid ethane at melting point. The grids were kept at liquid nitrogen temperatures until further processing.

**Automated preparation of cryo-lamellae.** Cryo-lamellae preparation was automated using an Aquilos 2 cryo FIB-SEM (Thermo Fisher). Grids were initially coated with GIS platinum for 20 s. Lamella positions were identified around the centre marker of the grids after automated tile acquisition with Maps (200× magnification, 5 kV, 13 pA). These positions were then transferred to the Cryo AutoTEM with a target lamella size of 14 µm length and approximately 170 nm thickness, with a 4 nm offset after final polishing, resulting in lamellae of approximately 180 nm thickness. Following the automated procedure, the lamellae were additionally sputter-coated with metallic platinum (30 mA, 1 kV,

for 1–2 s). The final lamellae were stored at liquid nitrogen temperatures until further imaging.

**Cryo-ET and tomogram reconstruction.** Cryo-electron tomograms were recorded on a Titan Krios G2 microscope (Thermo Fisher) operated at 300 kV, equipped with a FEG, BioQuantum imaging filter (Gatan) with a 20 eV energy width, and a K3 direct electron detector (Gatan). For automated acquisition of the tomograms we used SerialEM[43]. Recorded frames were motion corrected using Warp and dose filtered before tomogram reconstruction. Aligned frames from Warp were utilized for patch tracking or fiducial-based alignment of small platinum particles in IMOD for tomogram reconstruction. CTF correction was performed through phase flipping in IMOD. Tomograms for visualization and IsoNet training were reconstructed with binning 6 (11.90 Å pixel size) and an additional SIRT-like filter in IMOD, set to 50 iterations. A comprehensive overview of all acquired tomograms used in this study is provided in Supplementary Table 6.

**Missing wedge correction in IsoNet.** For missing wedge correction in IsoNet, we employed the tool at binning 6 (11.90 Å pixel size) for visualization and prohibitin quantifications[44]. IsoNet was trained separately using four tomograms from U2OS WT cells, four from Co7 WT cells, and four from rat hippocampal neurons. CTF deconvolution was not applied before masking. A mask was generated with density and s.d. percentages set at 60, and a z-crop of 0.05. Subtomograms (120 per input tomogram) were extracted, and the network underwent training for 35 iterations with 15 epochs each. Noise was introduced at iterations 16, 21, 26 and 31 with levels of 0.05, 0.1, 0.15 and 0.2, respectively. The resulting model after 35 iterations was used for predicting tomograms depicted in this study.

**Subtomogram averaging in Dynamo.** For subtomogram averaging, 37 tomograms from U2OS WT cells were utilized, and 817 particles were manually picked in Dynamo at a pixel size of 3.87 Å (ref. 24). An initial average was created by manually aligning 50 particles in dgallery. Subsequently, particles were globally aligned to the initial average using Dynamo's 'global search' pre-set. Randomization of particle orientations was performed using the 'dynamo_table_randomize_azimuth' function, and the resulting table was used for particle averaging. Pose-optimized particles were extracted in Warp at a 5 Å pixel size and aligned in Relion 4.0 through 3D auto-refinement, revealing an 11-fold symmetry of the prohibitin complex. All particles were then aligned to a C11-fold symmetrized template. The resulting star file from auto-refinement was used to re-extract particles in Warp at a 2.5 Å pixel size, followed by another round of 3D auto-refinement in Relion. Resolution estimation for all alignments was performed using the EMDB FSC server with two independent half maps generated from the Relion 4.0 3D auto-refinement[45]. Visualization of 3D models of the cryo-EM maps was conducted using ChimeraX[46].

#### Modelling

For the modelling of the prohibitin complex, we used the Alphafold predicted structures of PHB1 (Uniprot P35232) and PHB2 (Uniprot P50093), respectively. As starting point, the structure of Prohibitin 1 was choosen[47]. After placement of the globular domain in the density map, backbone rotations between residues 174–178 and 229–233 were made, analogous to the bacterial homologous complex, while a C11 symmetry was maintained using PyMOL[6,48]. Afterwards, the resultant model was fitted into the cryo-EM derived density map using the MD method from Flex-EM, with a cap-shift of 0.5 Å, 20 runs and 100 iterations per run[49]. Additionally, all secondary structure elements from the Alphafold model with high confidence in the predicted local distance difference test (pLDDT >70), were treated as rigid bodies. This provided the C-terminal structured part of the interacting helices (residues 176–253), while a new placement of more globular and transmembrane part (residues 1–174)

was made, since owing to their membrane interaction, a fitting in a density map including membranes is not anticipated. Using this arrangement, a homology model of an alternating complex of five PHB1 and six PHB2 was built using MODELLER 10.4 (ref. 50). Stereochemical restraints were added as well as helical restraints on the transmembrane domain for PHB2 (residues 19–39). To avoid interference of the longer disordered C- and N-terminal part of PHB2, these areas were redirected during modelling by setting a $z$-coordinate upper limit for residues 1–19 of 107 Å with a s.d. of 0.3 Å and a lower limit for residues 286–291 of 158 Å with the same s.d. The coordinates frame was identical to the cryo-EM derived map. From 300 models, after ranking by the discrete optimized protein energy score, two were selected and visually inspected[51]. All-atom systems including solvent and a lipid membrane were built using CHARMM-GUI as described previously[52,53]. The membrane consisted of POPC lipids in a hexagonal setup. After equilibration, 100 ns of a MD simulation were computed for each setup using GROMACS 2021.3 and the CHARMM36m force filed as described before[52,54,55]. Density maps were computed using GROmaps using a pixel size of 2.5 Å (ref. 56).

**Blender modelling of an average mitochondrion**
The mitochondrion was drawn to scale in Blender 4.0.2 using the following data: length, 1.5 µm; outer membrane diameter, 565 nm; inner membrane diameter, 540 nm; distance outer and inner membrane, 10 nm; crista diameter (without junctions), 480 nm; crista thickness, 10 nm; intercristae distance, 94 nm; thickness of all membranes, 5 nm.

**Identification and validation of crosslinked peptides**
Crosslink peptides were obtained from previously published studies[28–30] and were reformatted using custom R scripts for visualization using the web-based visualization tool xiVIEW[29–31,57]. We evaluated the crosslinks by mapping the residue pairs onto the prohibitin Protein Data Base structures using the ChimeraX bundle XMAS[33]. The crosslinks were mapped to the shortest distance while allowing a difference of 2 Å unless otherwise stated. For each residue pair, we extracted the Euclidean distance between the Cα atoms of the structure using XMAS and considered a crosslink in agreement with the structure if the Cα–Cα distance was smaller than or equal to 30 Å.

**Statistics and reproducibility**
All experiments were conducted at least in triplicate using independent sample preparations. Sample size was chosen on the basis of previous experience and standards in the field. For cryo-ET data sets the maximum available samples were used for data acquisition. For the data shown in Fig. 1a,b, cells were tested on a regular basis throughout the study in three independent experiments. For the data shown in Fig. 1e,f, 16 cells were used for mito-DK and 20 cells for PHB2–DK analysis. For the data shown in Fig. 3a, four independent experiments were used. For the data shown in Fig. 3b two independent experiments each were used. For the data shown in Fig. 3c, we used three independent experiments of PHB1 and PHB2 KDs. The Jarque–Bera test was used to assess the normal distribution of the data followed by the two-tailed unpaired $t$-test (T) to determine $P$ values in Excel 2016. These results are summarized in Supplementary Table 7. Data collection was not randomized and data collections and analysis were not performed blind to the conditions of the experiments.

**Reporting summary**
Further information on research design is available in the Nature Portfolio Reporting Summary linked to this article.

**Data availability**
Data generated in this study have been deposited at the EM Data Bank (www.ebi.ac.uk/emdb) and are available under accessions EMD-19459 and Protein Data Base ID 8RRH[58]. All other data supporting the findings of this study are available from the corresponding author on reasonable request. Source data are provided with this paper.

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

## Acknowledgements
We thank T. Koenen, N. Molitor, C. Dienemann and U. Steuerwald for excellent technical assistance and H. Hillen and T. Stephan for discussions. This work was supported by the Deutsche Forschungsgemeinschaft (DFG, German Research Foundation) under Germany's Excellence Strategy–EXC 2067/1-390729940 and by the European Research Council (ERCAdG no. 835102) (both to S.J.). This study was funded by the DFG-funded SFB 1286 (project A05 to S.J.), FOR 2848 (project Z01 to D.R. and S.J.).

## Author contributions
F.L., M.R., J.-N.D., M.L.V. and P.I. performed experimental work, analysed data and prepared figures. D.R. and D.W. carried out EM

experiments at RT. S.J. supervised the study. F.L., M.R., M.L.V. and S.J. wrote the manuscript with input from all authors.

## Funding

## Competing interests

The authors declare no conflicts of interest.

## Additional information

**Extended data** is available for this paper at https://doi.org/10.1038/s41556-025-01620-1.

**Correspondence and requests for materials** should be addressed to Stefan Jakobs.

**A**

PHB1-DK overexpression

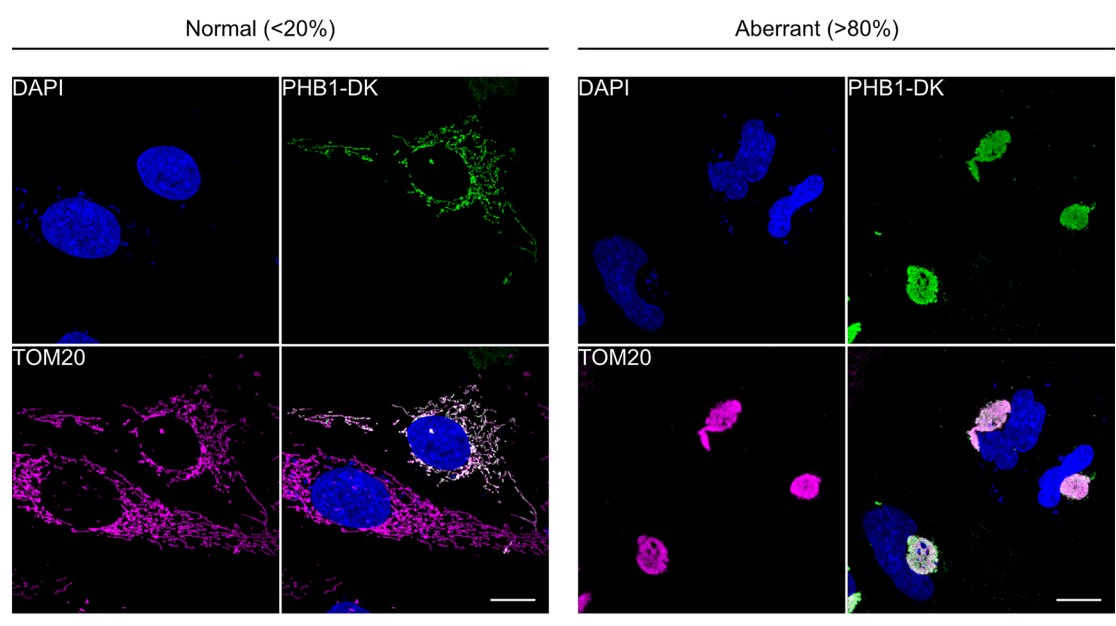

**B**

PHB2-DK overexpression

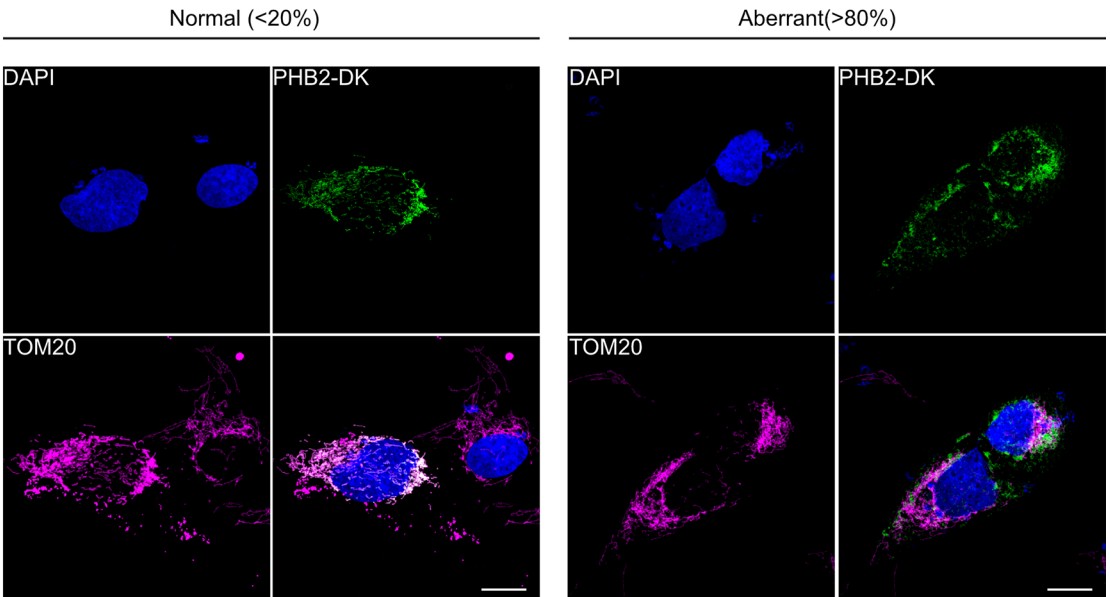

**Extended Data Fig. 1 | Overexpression of PHB1–DK and PHB2-DK in human cells.** U2OS cells were transfected with a plasmid encoding PHB1-DK (**a**) or PHB2-DK (**b**). For each construct, three separate transfection experiments were performed (n = 3); each time more than 100 fluorescent cells were analysed, and mitochondrial morphology was classified as "normal" or "aberrant". Scale bars, 10 µm.

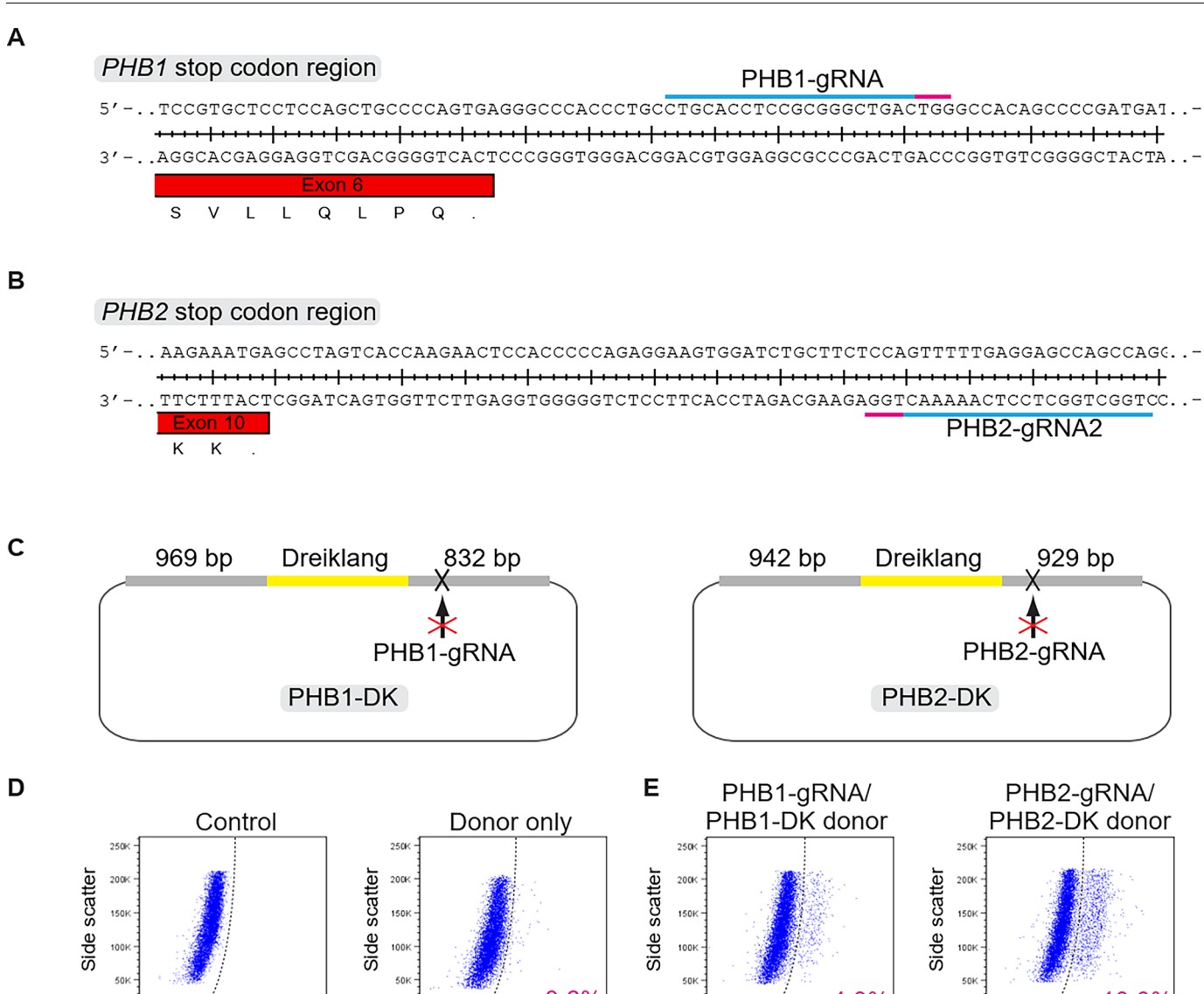

**Extended Data Fig. 2 | Guide RNAs and donor plasmids for C-terminal tagging of human prohibitins.** (**a**, **b**) Two gRNAs were designed for targeting the stop codon region of human PHB1 (**a**) or PHB2 (**b**), respectively. Light blue: gRNA binding site; Magenta, protospacer adjacent motif (PAM) site. (**c**) Donor plasmids encoding the fluorescent protein Dreiklang (DK) flanked by gene-specific homology arms containing point mutations in the PAM sites rendering the respective construct resistant to Cas9-mediated degradation. (**d**, **e**) FACS analysis of U2OS cells. Untransfected (control) and donor plasmid only (donor only) transfected U2OS cells were used as a negative control to set FACS gates (**d**). Co-transfection of PHB1-gRNA1 or PHB2-gRNA2 with Cas9-resistant PHB1-DK or PHB2-DK donor plasmids (**e**). The mean fraction of DK+ cells of three independent experiments is shown in magenta.

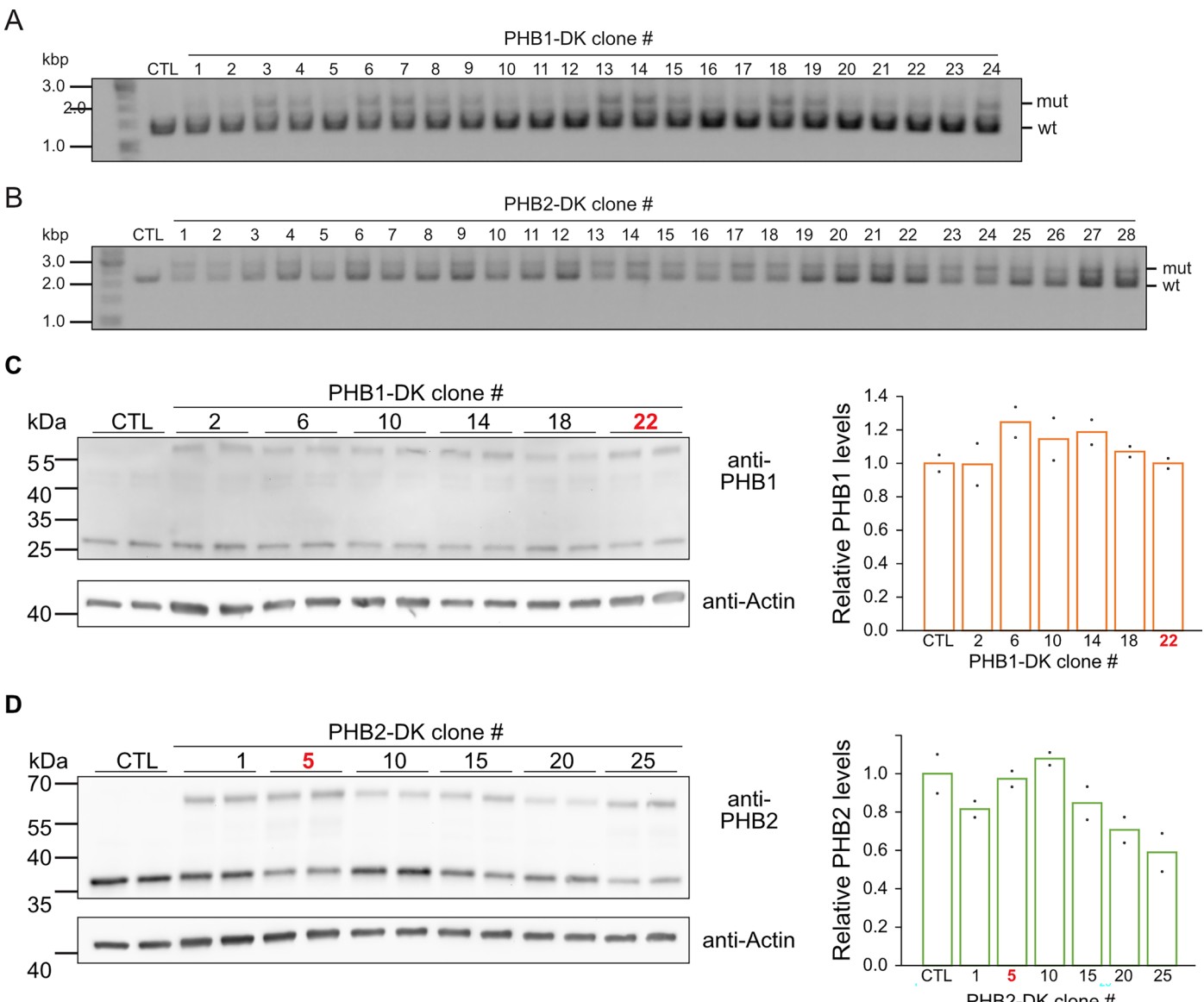

**Extended Data Fig. 3 | Genotyping and expression levels in PHB1-DK and PHB2-DK knock-in clones.** (**a**, **b**) PCR analysis of 24 PHB1-DK clones (**a**) and 28 PHB2-DK clones (**b**) obtained after single cell selection using FACS and clonal expansion. CTL, parental, non-edited control cell line. (**c**, **d**) Protein expression level analysis in five PHB1-DK (**c**) and five PHB2-DK clones (**d**). Whole cell extracts of clones were analysed via immunoblotting using antibodies against PHB1 (**c**)

or PHB2 (**d**), respectively. Wildtype extract (CTL) was loaded as a reference and actin was detected as internal loading control. Band intensities were quantified, PHB1 (**c**) or PHB2 (**d**) expression levels corrected for variations in loaded amounts and normalized to the PHB1 (**c**) or PHB2 (**d**) expression level in wildtype (CTL) cells. Clone numbers highlighted in red indicate the respective clone chosen for subsequent analysis. Source blots are available.

**A**

PHB1-DK clone, PHB1 locus

PHB1-Exon 6                    PHB1-gRNA    ▼    PAM
Reference  AGCTGCCCCAGTGAGGGCCCACCCTGCCTGCACCTCCGCGGGCTGACTGG
PHB1-DK    AGCTGCCCCAGTGAGGGCCCACCCTGCCTGCACCTCCGCGGGCTGACTGG

DK                            PHB1-gRNA    ▼    PAM
Reference  ACAAGTAAGGTACCGGGCCCACCCTGCCTGCACCTCCGCGGGCTGACTGG
PHB1-DK    ACAAGTAAGGTACCGGGCCCACCCTGCCTGCACCTCCGCGGGCTGACTAA

**B**

PHB2-DK clone, PHB2 locus

PHB2-Exon 10                 PAM    ▼    PHB2-gRNA
Reference     AGGGTAAGAAATGAN$_{37}$ATCTGCTTCTGGAGTTTTTGAGGAGCCAGCCAGGGG
PHB2-DK #1    AGGGTAAGAAATGAN$_{37}$ATCTGCTTCTGGAGTTTTTGAGGAGCCAGCCAGGGG
PHB2-DK #2    AGGGTAAGAAATGAN$_{37}$ATCTGCTTCTGGAGTT--------------CAGGGG
PHB2-DK #3    AGGGTAAGAAATGAN$_{37}$ATCTGCTTCTGGAGTT--------------------

DK                            PAM    ▼    PHB2-gRNA
Reference     ACAAGTAAGGTACCN$_{37}$ATCTGCTTCTGGAGTTTTTGAGGAGCCAGCCAGGGG
PHB2-DK       ACAAGTAAGGTACCN$_{37}$ATCTGCTTCTTTAGTTTTTGAGGAGCCAGCCAGGGG

**C**

PHB1-DK clone, top 3 off-target (OT) sites

OT1    ▼    PAM
Reference  GAGGGCCCACCCTGCCTGCACCTCCGCAGGCTGACTGGGCCACAGCCCCA
PHB1-DK    GAGGGCCCACCCTGCCTGCACCTCCGCAGGCTGACTGGGCCACAGCCCCA

OT2    ▼    PAM
Reference  GAGGGCCCACACGGCCTCCACTTCCGCGGGCTGACCGGGCCACAGCCCCG
PHB1-DK    GAGGGCCCACACGGCCTCCACTTCCGCGGGCTGACCGGGCCACAGCCCCG

OT3    ▼    PAM
Reference  GAGGGCCCATCCTGCCTGCACCGCCGTGGGCTGACTGGGGCACAGCTCCG
PHB1-DK    GAGGGCCCATCCTGCCTGCACCGCCGTGGGCTGACTGGGGCACAGCTCCG

**D**

PHB2-DK clone, top 3 off-target (OT) sites

OT1    ▼    PAM
Reference  TGGGATCCCTACAGCCTTGCTGGCTCTTCAAAAACAAGCCCATCACTTCA
PHB2-DK    TGGGATCCCTACAGCCTTGCTGGCTCTTCAAAAACAAGCCCATCACTTCA

OT2    ▼    PAM
Reference  GTCGGGGGTAGGGCTCAGGCTGGCTCCTCCAAAACAAGGGACGCTGACAG
PHB2-DK    GTCGGGGGTAGGGCTCAGGCTGGCTCCTCCAAAACAAGGGACGCTGACAG

OT3    ▼    PAM
Reference  CCTCTGGACAGGCTCCTGCCTCACTCCTCAAAAACGGGACAAAGGGCCAG
PHB2-DK    CCTCTGGACAGGCTCCTGCCTCACTCCTCAAAAACGGGACAAAGGGCCAG

**Extended Data Fig. 4 | See next page for caption.**

**Extended Data Fig. 4 | Sanger sequencing of on-target and predicted possible off-target sites in PHB1-DK and PHB2-DK knock-in clones. (a)** Sanger sequencing of the selected heterozygous PHB1-DK clone revealed that the untagged PHB1 allele is unmodified (top) and that the tagged PHB1-DK allele contains the expected mutations introduced into the donor plasmid (bottom). (**b**) Sequencing of the selected heterozygous PHB2-DK clone revealed that the untagged PHB2 alleles contain three DNA sequence variants (top): unmodified

sequence (#1), a 14 bp deletion (#2) or a 20 bp deletion (#3). The tagged PHB2-DK allele contains the expected mutations introduced into the donor plasmid (bottom). (**c, d**) Absence of mutations at the predicted top three possible off-target sites for the respective gRNA used to generate PHB1-DK (**c**) and PHB2-DK (**d**) suggests precise gene editing. For all experiments, the respective DNA regions were amplified from genomic DNA, subcloned and sequenced using Sanger sequencing at a depth of at least 3x.

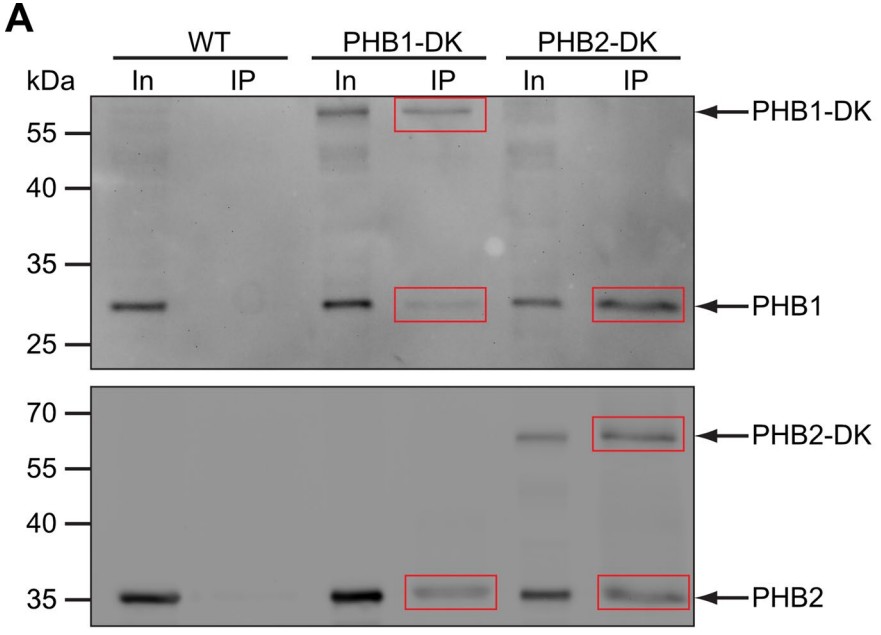

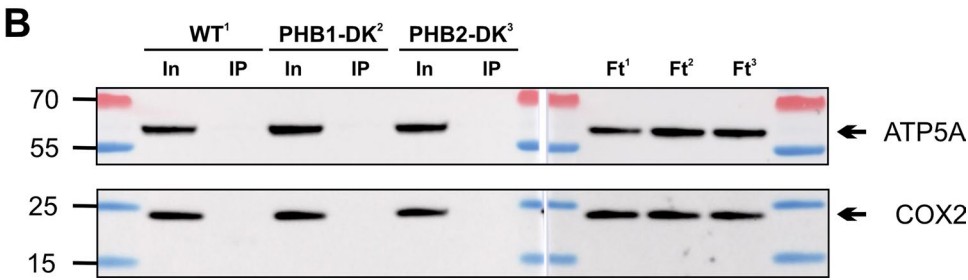

**Extended Data Fig. 5 | Co-immunoprecipitation of prohibitins.**
Co-immunoprecipitation of prohibitins from isolated mitochondria. Mitochondrial lysates from PHB-DK1 or PHB-DK2 knock-in cells were immunoprecipitated using Dreiklang as the bait protein. (**a**) Antibody detection; upper blot: PHB1, lower blot: PHB2. (**b**) Antibody detection; upper blot: ATP5A, lower blot: COX2. FT1: flow through of wildtype CoIP, FT2: flow through of PHB1-DK CoIP, FT3: flow through of PHB2-DK CoIP. 10 µg of sample was loaded for all samples of Input and flow through, respectively. Source blots are available.

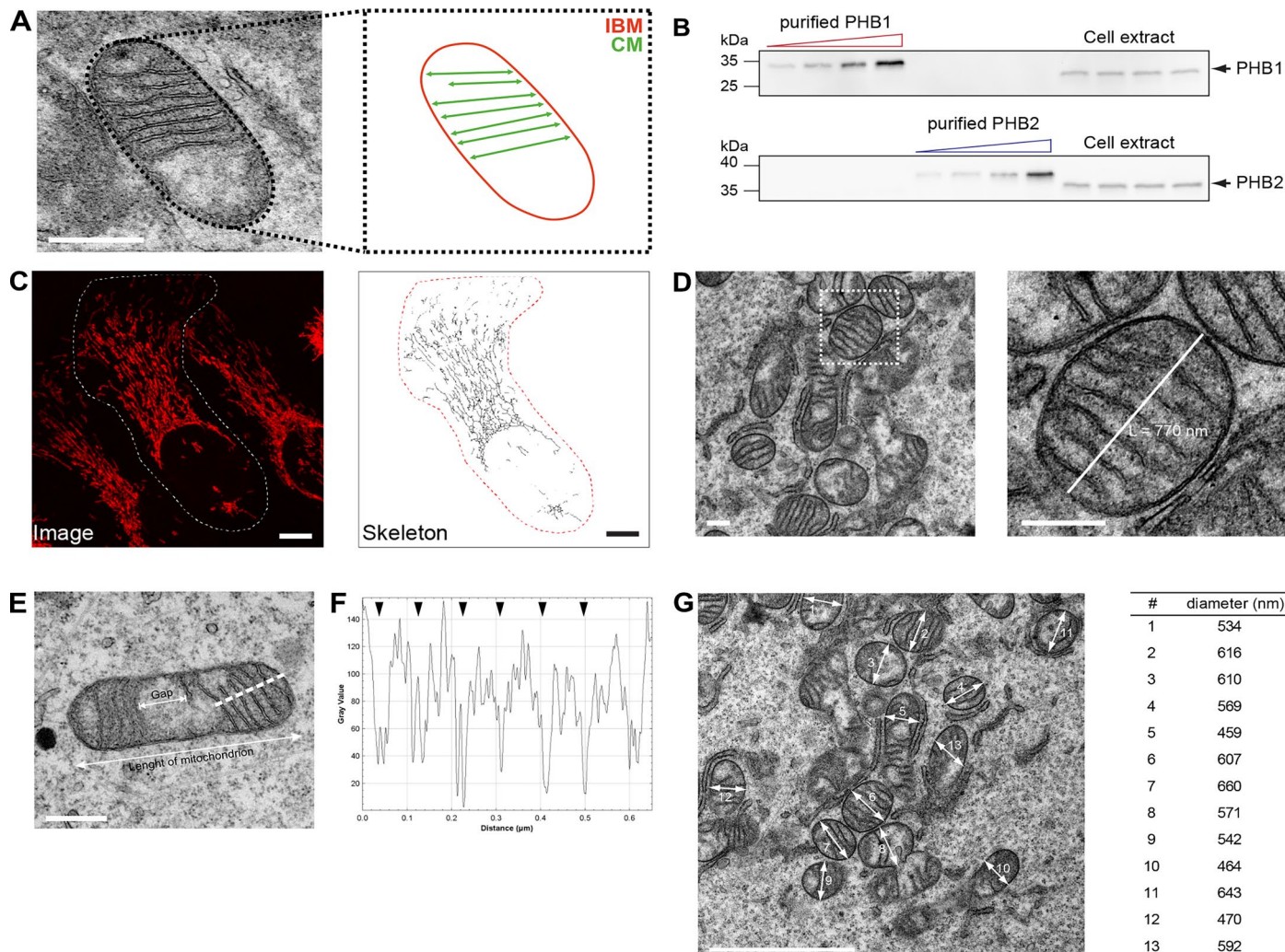

**Extended Data Fig. 6 | Quantitative analysis of PHB complexes.**
(**a**) Representative TEM micrograph of a U2OS mitochondrion. The IBM to CM ratio was determined by measuring the total IBM length (red) and summing the lengths of all cristae, multiplied by 2 due to the dual-membrane composition of a crista. The U2OS wt cells exhibited an IBM to CM ratio of 0.64 to 1 (n = 32 mitochondria). (**b**) Quantitative analysis of prohibitin abundance in cells. Increasing amounts of purified PHB1 and PHB2 were used as a reference to estimate the total number of prohibitin molecules in U2OS cells. Purified prohibitins and U2OS cell extracts were loaded on the same gel and blotted for antibody-based detection of PHB1 (top) or PHB2 (bottom). Note that purified PHB1 and PHB2 run at a higher molecular weight than endogenous prohibitins due to the His-tag and linker sequence. (**c**) Determination of total mitochondrial network length by image analysis. U2OS cells stained with Mitotracker Deep Red FM revealed a mitochondria network length per cell of 1082 ± 360 μm

(mean ± SD, n = 34 cells). A representative image is shown with a network length of 1029 μm. (**d**) Representative TEM micrograph of a U2OS mitochondrion. Average distance between crista membranes determined from individual mitochondria was determined as 74.1 ± 15.1 nm, SD, n = 36 mitochondria. (**e**) Representative TEM micrograph of a U2OS mitochondrion used to determine the total mitochondrial length and size of voids between groups of cristae. Measurements revealed an average gap size of 21 ± 8% of the mitochondrial tubules (n = 98 mitochondria). (**f**) Line profile along the white dashed line in (**e**). This mitochondrion shows an average inter-crista distance of approximately 79 nm. Cristae are indicated with black arrowheads. (**g**) Representative EM micrograph of U2OS cells. Data were used to measure the diameter of mitochondria. Scale bar in A, 500 nm. Scale bar in C, 10 μm. Scale bar in D, E 300 nm. Scale bar in G, 2 μm. Source data are available.

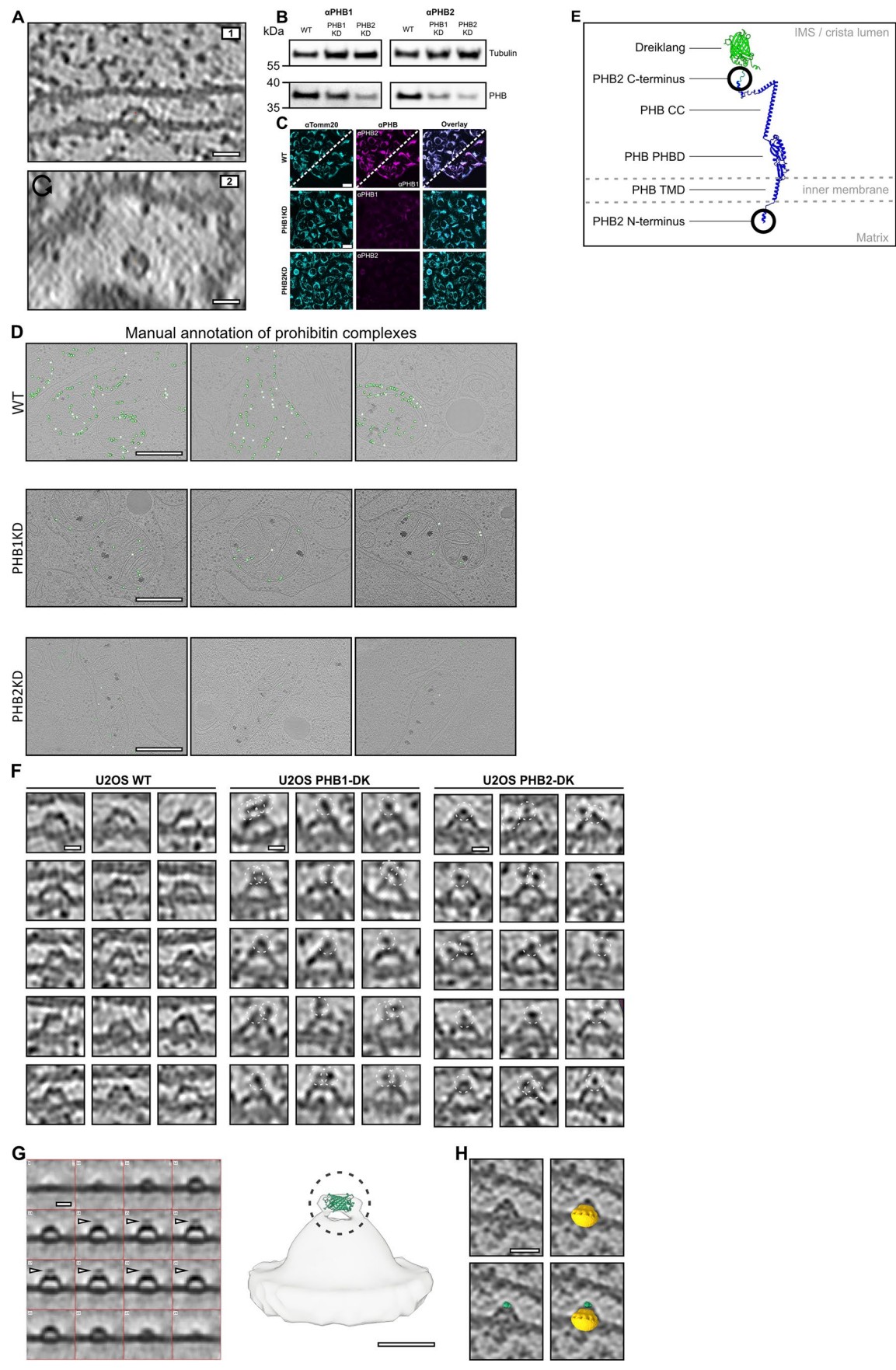

**Extended Data Fig. 7 | See next page for caption.**

**Extended Data Fig. 7 | Prohibitin structures. (a)** IMOD slicer view of a representative particle recorded on mitochondria of wildtype U2OS cells. 1: Side view shows a convex profile of the particle, tilted to 10° (x-rotation). 2: Top view exhibits a ring-like appearance of the same particle, tilted to 90° (x-rotation) with respect to (1). **(b)** Western blot analysis. RNAi-mediated knockdown of PHB1 and PHB2 showed that depletion of one prohibitin leads also to the reduction of the other paralog. **(c)** Immunofluorescence imaging of PHB1KD and PHB2KD cells using an antibody against PHB1 and PHB2, respectively. Mitochondrial network labelled with Tomm20 antibodies (cyan). **(d)** Representative central tomographic slices of U2OS cells. Prohibitin structures were marked with green spheres. PHB1KD and PHB2KD cells exhibit reduced prohibitin abundances compared to wildtype cells. **(e)** AlphaFold-predicted structure of the PHB2-Dreiklang fusion protein (PHB2 in blue, Dreiklang in green). **(f)** Central slices of tomograms of individual prohibitin complexes. PHB2-DK cells show additional densities at the top of the structures, indicating PHB2 molecules fused to Dreiklang. **(g)** Left: Subtomogram average of PHB-DK particles (U2OS PHB2-DK cells). STA confirms the presence of an additional density at the bell top, indicating the presence of DK (white arrowheads). Right: Isosurface rendering in ChimeraX (transparent) with the AlphaFold-predicted structure of Dreiklang fitted into the density. **(h)** Central tomographic slice of a prohibitin assembly. Image overlay with a volume representation of the prohibitin cryo-EM map (orange) and a Dreiklang barrel (green). The additional density at the bell top matches the size of the Dreiklang barrel. Scale bars in A, 20 nm. Scale bars in C, 50 μm. Scale bars in D, 500 nm. Scale bars in F, G, 10 nm. Scale bar in H, 20 nm. Source data and blots are available.

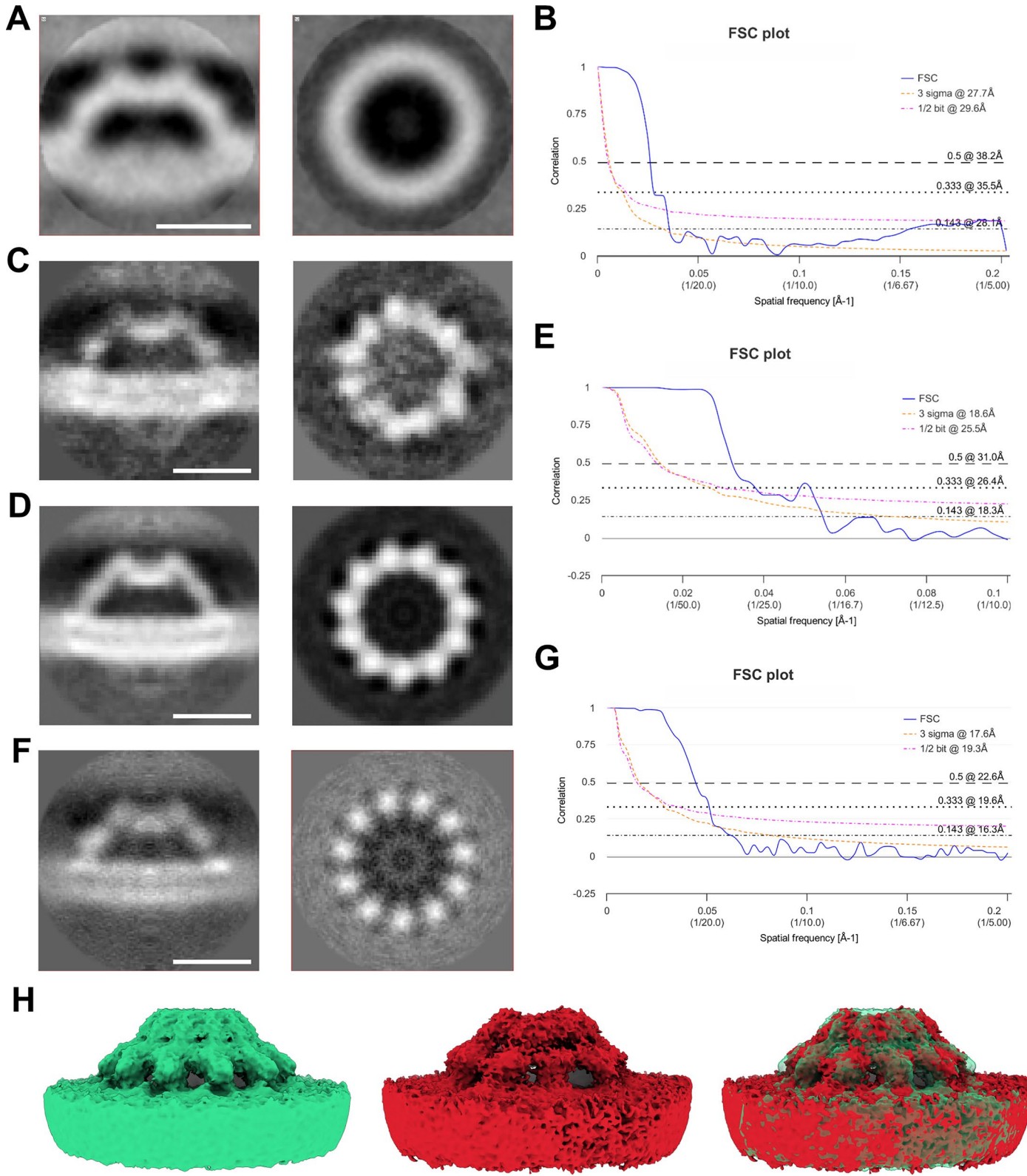

**Extended Data Fig. 8 | Subtomogram averaging of prohibitin complexes.**
(**a**) Initial average of the prohibitin complex obtained through subtomogram averaging in Dynamo (3.87 Å pixel size). Left: X-view of the Cryo-EM map displaying a convex structure with a hollow core. Right: Z-view showing a ring-like structure. (**b**) Fourier shell correlation. A resolution of 28.1 Å (0.143 criterion) was obtained for the average presented in (**a**). (**c**) Average after initial alignment of subvolumes with 5 Å pixel size. The cryo-EM map suggests an underlying C11 symmetry of the prohibitin complex. (**d**) Subtomogram average of volumes with 5 Å pixel size aligned with C11 symmetry. Left: X-view of the average.

Right: Top-view of the average. (**e**) Fourier shell correlation. A resolution of 18.3 Å (0.143 criterion) was obtained for the average presented in (**d**). (**f**) Subtomogram average of volumes with 2.5 Å pixel size aligned with C11 symmetry. Left: X-view of the average. Right: Top-view of the average. (**g**) Fourier shell correlation. A resolution of 16.3 Å (0.143 criterion) was obtained for the average presented in (**f**). (**h**) Isosurface representations. The final cryo-EM map with C11 symmetry (green, left), alignment without symmetry (red, middle), and an overlay of both maps (right). Scale bars, 10 nm.

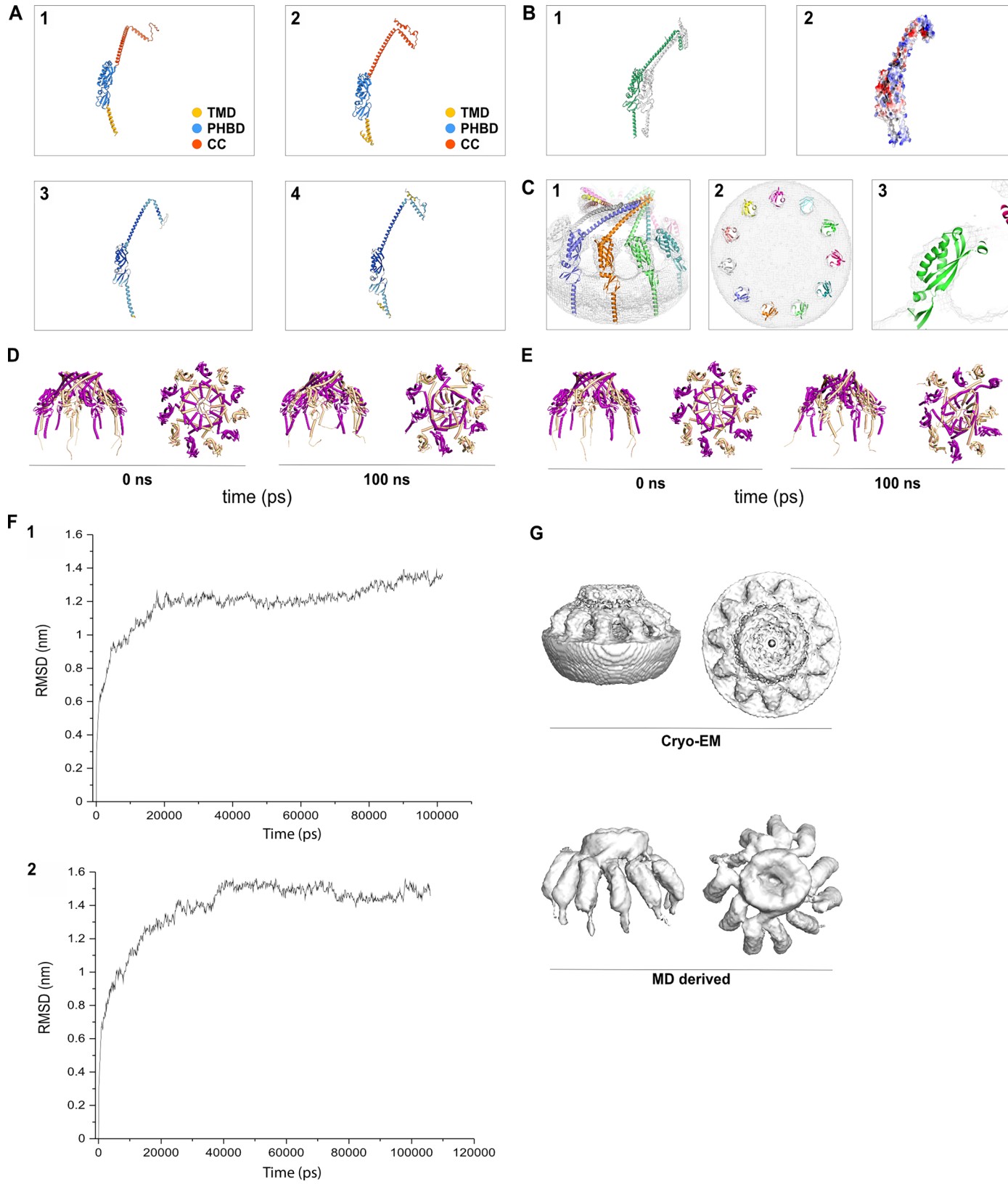

**Extended Data Fig. 9 | See next page for caption.**

**Extended Data Fig. 9 | Molecular diffusion of the prohibitin complex model.**
(**a**) 1-2: AlphaFold-predicted structures of the human prohibitins. Prohibitin 1
(PHB1, Uniprot P35232) and prohibitin 2 (PHB2, Uniprot Q99623) showing key
domains – transmembrane (yellow), PHB (blue), and coiled-coil (orange). 3-4:
AlphaFold-predicted structures coloured according to the pLDDT (dark blue:
pLDDT > 90, light blue: 90 > pLDDT > 70, yellow: 70 > pLDDT > 50, orange: pLDDT
< 50). (**b**) 1-2: AlphaFold-predicted model of the PHB1 (green) and PHB2 (white)
dimer. Structural model (1). Electrostatic potential map (2). (**c**) 1–3: Placement of
individual prohibitin molecules in the final cryo-EM map. The densities only fit a
single prohibitin molecule each. Side view (1). Top view (2). Detailed view on the
large density at the lipid bilayer fitting a single PHB domain (3).

(**d**, **e**) Two models for the prohibitin complex used in molecular dynamics (MD)
simulations (PHB1 in magenta, PHB2 in green). Preservation of the complex after
100 ns simulations is observed, with increased stability at individual coiled-coil
domains and flexibility at N-terminus and PHB domains. (**f**) RMSD curves of the
MD simulations. RMSD of the model shown in D converged to ~1.4 nm (1). RMSD
of the model shown in E converged to ~1.6 nm (2). (**g**) Isosurface representations.
Comparison of the cryo-EM-derived map and an MD-derived map after 100 ns,
excluding lipids for a protein-only comparison. The MD model maintains
structural similarities with the initial model, indicating overall preservation
during MD.

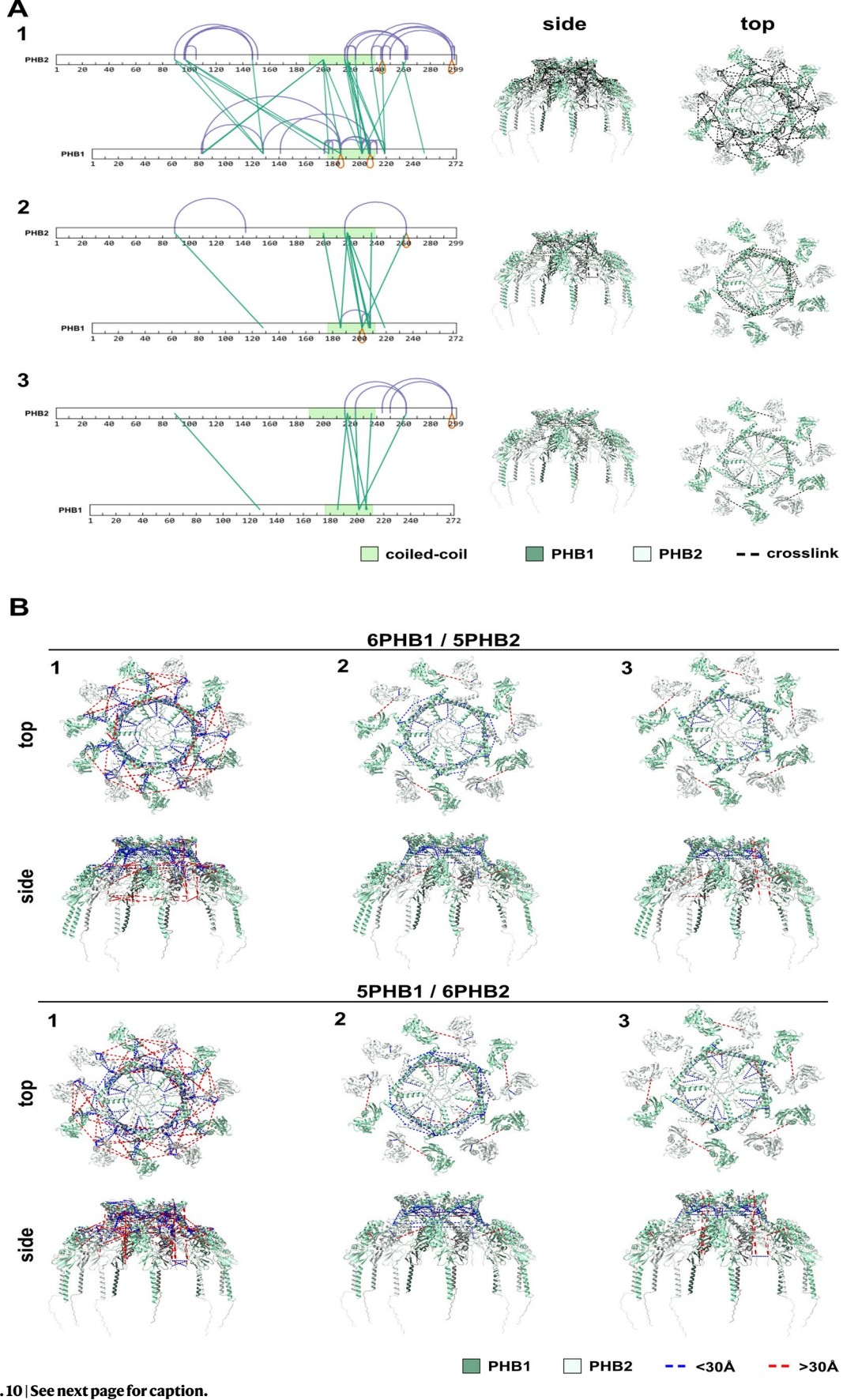

**Extended Data Fig. 10 | See next page for caption.**

**Extended Data Fig. 10 | Crosslinking and mass spectrometry analysis of the prohibitin complex. (a)** 1–3: Crosslinking mass spectrometry analyses of the mouse and human prohibitin heteromeric complex. Schematic representation of the distribution of the intra-subunit crosslinks (purple), inter-subunit crosslinks (green), and self-crosslinks (orange) identified in three separate studies (1: Ryl et al., 2020, 2: Schweppe et al., 2017, and 3: Liu et al., 2018). The predicted prohibitin coiled coils are shown in green (left). Crosslinks identified were mapped to the PHB1(5)PHB2(6) heteromeric structure(right). **(b)** 1–3: Crosslinked sites mapped to the prohibitin heteromeric structures. Crosslinks identified in three separate studies (1: Ryl et al., 2020, 2: Schweppe et al., 2017, and 3: Liu et al., 2018) were mapped to the PHB1(6)PHB2(5) (top) or PHB1(5)PHB2(6) (bottom) heteromeric structures. Crosslinks that validate the structure (<30 Å) are shown in blue while unsatisfied crosslinks (>30 Å) are shown in red.

# Reporting Summary

## Statistics

For all statistical analyses, confirm that the following items are present in the figure legend, table legend, main text, or Methods section.

| n/a | Confirmed | |
|---|---|---|
| ☐ | ☒ | The exact sample size (*n*) for each experimental group/condition, given as a discrete number and unit of measurement |
| ☐ | ☒ | A statement on whether measurements were taken from distinct samples or whether the same sample was measured repeatedly |
| ☐ | ☒ | The statistical test(s) used AND whether they are one- or two-sided *Only common tests should be described solely by name; describe more complex techniques in the Methods section.* |
| ☒ | ☐ | A description of all covariates tested |
| ☐ | ☒ | A description of any assumptions or corrections, such as tests of normality and adjustment for multiple comparisons |
| ☐ | ☒ | A full description of the statistical parameters including central tendency (e.g. means) or other basic estimates (e.g. regression coefficient) AND variation (e.g. standard deviation) or associated estimates of uncertainty (e.g. confidence intervals) |
| ☐ | ☒ | For null hypothesis testing, the test statistic (e.g. *F*, *t*, *r*) with confidence intervals, effect sizes, degrees of freedom and *P* value noted *Give P values as exact values whenever suitable.* |
| ☒ | ☐ | For Bayesian analysis, information on the choice of priors and Markov chain Monte Carlo settings |
| ☒ | ☐ | For hierarchical and complex designs, identification of the appropriate level for tests and full reporting of outcomes |
| ☒ | ☐ | Estimates of effect sizes (e.g. Cohen's *d*, Pearson's *r*), indicating how they were calculated |

*Our web collection on statistics for biologists contains articles on many of the points above.*

## Software and code

Policy information about availability of computer code

| Data collection | SerialEM 4.0, Cryo AutoTEM, LAS AF, BD FACSDiva, TVIPS EM-Menu |
|---|---|
| Data analysis | Relion 4.0, Dynamo 1.1.532, IMOD 4.11, MODELLER 10.4, PyMOL 2.5, GROMACS, GROmaps, IsoNet 0.2, Excel 2016, Blender 4.2 |

For manuscripts utilizing custom algorithms or software that are central to the research but not yet described in published literature, software must be made available to editors and reviewers. We strongly encourage code deposition in a community repository (e.g. GitHub). See the Nature Portfolio guidelines for submitting code & software for further information.

## Data

Policy information about availability of data

All manuscripts must include a data availability statement. This statement should provide the following information, where applicable:
- Accession codes, unique identifiers, or web links for publicly available datasets
- A description of any restrictions on data availability
- For clinical datasets or third party data, please ensure that the statement adheres to our policy

Data will be available under accessions EMD-19459 (EMDB) and PDB ID 8RRH (PDB). The mass spectrometry data used for analysis in this study was cited at multiple occasions for the readers to follow. Source data have been provided in Source Data. All other data supporting the findings of this study are available from the corresponding author on reasonable request.

# Research involving human participants, their data, or biological material

Policy information about studies with human participants or human data. See also policy information about sex, gender (identity/presentation), and sexual orientation and race, ethnicity and racism.

| | |
|---|---|
| Reporting on sex and gender | N/A |
| Reporting on race, ethnicity, or other socially relevant groupings | N/A |
| Population characteristics | N/A |
| Recruitment | N/A |
| Ethics oversight | N/A |

Note that full information on the approval of the study protocol must also be provided in the manuscript.

# Field-specific reporting

Please select the one below that is the best fit for your research. If you are not sure, read the appropriate sections before making your selection.

☒ Life sciences ☐ Behavioural & social sciences ☐ Ecological, evolutionary & environmental sciences

For a reference copy of the document with all sections, see [nature.com/documents/nr-reporting-summary-flat.pdf](http://nature.com/documents/nr-reporting-summary-flat.pdf)

# Life sciences study design

All studies must disclose on these points even when the disclosure is negative.

| | |
|---|---|
| Sample size | All experiments were conducted at least in triplicate using independent sample preparations. Sample size were chosen based on previous experience and standards in the field. For cryoET data sets the maximum available samples were used for data acquisition. |
| Data exclusions | No data was excluded to ensure a comprehensive representation of our findings. |
| Replication | All attempts to replication were successful. Samples were replicated in at least three independent wells. |
| Randomization | In this study, controlling for covariates was not directly feasible due to the structure of the experimental design. Specifically, the samples were systematically arranged on culture plates and cryo EM grids without employing randomization thereby ensuring equal representation of all conditions. This specific organization minimized the impact of potential unforeseen variables resulting in reliable and consistent data for analysis. |
| Blinding | Blinding was not possible in this study, as one individual conducted most of the light microscopy analysis and another handled the cryo-ET analysis. However, to minimize potential bias additional authors participated in validating the analysis and thereby ensuring the objectivity of the results. |

# Behavioural & social sciences study design

All studies must disclose on these points even when the disclosure is negative.

| | |
|---|---|
| Study description | *Briefly describe the study type including whether data are quantitative, qualitative, or mixed-methods (e.g. qualitative cross-sectional, quantitative experimental, mixed-methods case study).* |
| Research sample | *State the research sample (e.g. Harvard university undergraduates, villagers in rural India) and provide relevant demographic information (e.g. age, sex) and indicate whether the sample is representative. Provide a rationale for the study sample chosen. For studies involving existing datasets, please describe the dataset and source.* |
| Sampling strategy | *Describe the sampling procedure (e.g. random, snowball, stratified, convenience). Describe the statistical methods that were used to predetermine sample size OR if no sample-size calculation was performed, describe how sample sizes were chosen and provide a rationale for why these sample sizes are sufficient. For qualitative data, please indicate whether data saturation was considered, and what criteria were used to decide that no further sampling was needed.* |
| Data collection | *Provide details about the data collection procedure, including the instruments or devices used to record the data (e.g. pen and paper, computer, eye tracker, video or audio equipment) whether anyone was present besides the participant(s) and the researcher, and whether the researcher was blind to experimental condition and/or the study hypothesis during data collection.* |

| Timing | *Indicate the start and stop dates of data collection. If there is a gap between collection periods, state the dates for each sample cohort.* |
| --- | --- |
| Data exclusions | *If no data were excluded from the analyses, state so OR if data were excluded, provide the exact number of exclusions and the rationale behind them, indicating whether exclusion criteria were pre-established.* |
| Non-participation | *State how many participants dropped out/declined participation and the reason(s) given OR provide response rate OR state that no participants dropped out/declined participation.* |
| Randomization | *If participants were not allocated into experimental groups, state so OR describe how participants were allocated to groups, and if allocation was not random, describe how covariates were controlled.* |

# Ecological, evolutionary & environmental sciences study design

All studies must disclose on these points even when the disclosure is negative.

| Study description | *Briefly describe the study. For quantitative data include treatment factors and interactions, design structure (e.g. factorial, nested, hierarchical), nature and number of experimental units and replicates.* |
| --- | --- |
| Research sample | *Describe the research sample (e.g. a group of tagged Passer domesticus, all Stenocereus thurberi within Organ Pipe Cactus National Monument), and provide a rationale for the sample choice. When relevant, describe the organism taxa, source, sex, age range and any manipulations. State what population the sample is meant to represent when applicable. For studies involving existing datasets, describe the data and its source.* |
| Sampling strategy | *Note the sampling procedure. Describe the statistical methods that were used to predetermine sample size OR if no sample-size calculation was performed, describe how sample sizes were chosen and provide a rationale for why these sample sizes are sufficient.* |
| Data collection | *Describe the data collection procedure, including who recorded the data and how.* |
| Timing and spatial scale | *Indicate the start and stop dates of data collection, noting the frequency and periodicity of sampling and providing a rationale for these choices. If there is a gap between collection periods, state the dates for each sample cohort. Specify the spatial scale from which the data are taken* |
| Data exclusions | *If no data were excluded from the analyses, state so OR if data were excluded, describe the exclusions and the rationale behind them, indicating whether exclusion criteria were pre-established.* |
| Reproducibility | *Describe the measures taken to verify the reproducibility of experimental findings. For each experiment, note whether any attempts to repeat the experiment failed OR state that all attempts to repeat the experiment were successful.* |
| Randomization | *Describe how samples/organisms/participants were allocated into groups. If allocation was not random, describe how covariates were controlled. If this is not relevant to your study, explain why.* |
| Blinding | *Describe the extent of blinding used during data acquisition and analysis. If blinding was not possible, describe why OR explain why blinding was not relevant to your study.* |

Did the study involve field work?  ☐ Yes  ☐ No

# Field work, collection and transport

| Field conditions | *Describe the study conditions for field work, providing relevant parameters (e.g. temperature, rainfall).* |
| --- | --- |
| Location | *State the location of the sampling or experiment, providing relevant parameters (e.g. latitude and longitude, elevation, water depth).* |
| Access & import/export | *Describe the efforts you have made to access habitats and to collect and import/export your samples in a responsible manner and in compliance with local, national and international laws, noting any permits that were obtained (give the name of the issuing authority, the date of issue, and any identifying information).* |
| Disturbance | *Describe any disturbance caused by the study and how it was minimized.* |

# Reporting for specific materials, systems and methods

We require information from authors about some types of materials, experimental systems and methods used in many studies. Here, indicate whether each material, system or method listed is relevant to your study. If you are not sure if a list item applies to your research, read the appropriate section before selecting a response.

## Materials & experimental systems

| n/a | Involved in the study |
|---|---|
| ☐ | ☒ Antibodies |
| ☐ | ☒ Eukaryotic cell lines |
| ☒ | ☐ Palaeontology and archaeology |
| ☐ | ☒ Animals and other organisms |
| ☒ | ☐ Clinical data |
| ☒ | ☐ Dual use research of concern |
| ☒ | ☐ Plants |

## Methods

| n/a | Involved in the study |
|---|---|
| ☒ | ☐ ChIP-seq |
| ☐ | ☒ Flow cytometry |
| ☒ | ☐ MRI-based neuroimaging |

## Antibodies

| | |
|---|---|
| Antibodies used | anti-PHB1 (EP2803Y, 1:2000, Abcam, Cambridge, UK), anti-PHB2 (EPR14523, 1:5000, Abcam, Cambridge, UK), anti-GFP (JL-8, 1:3000, Clontech, Saint-Germain-en-Laye, France), anti-Actin (AC74, 1:3000, Sigma-Aldrich, St. Louis, MO, USA), HRP-conjugated anti-rabbit or anti-mouse secondary antibodies (Dianova, Hamburg, Germany), rabbit anti-GFP (ab290, Abcam, Cambridge, UK), sheep anti-mouse and goat anti-rabbit (all Dianova, 1:5000, Hamburg, Germany) coupled to KK114 or Alexa 594 (Atto-Tec, Siegen, Germany), anti-COX2 (ab203912, 1:500, Abcam, Cambridge, UK), anti-ATP5A (ab14748, 1:1000, Abcam, Cambridge, UK), anti-tubulin (ab15246, 1:1000, Abcam, Cambridge, UK), anti-ESR1 (sc-8005, 1:Santa Cruz Biotechnology, Texas, USA). |
| Validation | All antibodies were validated by us on KO/or knockdown cells. <br> In addition: <br> anti-PHB1 validated by Abcam (https://www.abcam.com/en-us/products/primary-antibodies/prohibitin-antibody-ep2803y-ab75766), anti-PHB2 validated by Abcam (https://www.abcam.com/en-us/products/primary-antibodies/rea-antibody-epr14523-ab182139), anti-GFP validated by Clontech (https://www.takarabio.com/products/antibodies-and-elisa/fluorescent-protein-antibodies/green-fluorescent-protein-antibodies), anti-Actin validated by Sigma Aldrich (https://www.sigmaaldrich.com/DE/de/product/sigma/a2228#product-documentation), rabbit anti-GFP validated by Abcam (https://www.abcam.com/en-us/products/primary-antibodies/gfp-antibody-ab290), anti-COX2 validated by Abcam (https://www.abcam.com/en-us/products/primary-antibodies/mtco1-antibody-epr19628-ab203912), anti-ATP5A validated by Abcam (https://www.abcam.com/en-us/products/primary-antibodies/atp5a-antibody-15h4c4-mitochondrial-marker-ab14748), anti-tubulin validated by Abcam (https://www.abcam.com/en-us/products/primary-antibodies/alpha-tubulin-antibody-microtubule-marker-ab15246), anti-ESR1 validated by Santa Cruz Biotechnology (https://www.scbt.com/de/p/estrogen-receptor-alpha-antibody-d-12). |

## Eukaryotic cell lines

Policy information about cell lines and Sex and Gender in Research

| | |
|---|---|
| Cell line source(s) | U2OS (Cat. No.: HTB-96, ATCC) Cos-7 (Cat. No.: 87021302, Sigma Aldrich) |
| Authentication | All by supplier: STR profiling (U2OS), karyotyping (U2OS), morphology (U2OS, Cos-7) |
| Mycoplasma contamination | negative; cells were tested regularly using extraction-free PCR. |
| Commonly misidentified lines (See ICLAC register) | none |

## Palaeontology and Archaeology

| | |
|---|---|
| Specimen provenance | *Provide provenance information for specimens and describe permits that were obtained for the work (including the name of the issuing authority, the date of issue, and any identifying information). Permits should encompass collection and, where applicable, export.* |
| Specimen deposition | *Indicate where the specimens have been deposited to permit free access by other researchers.* |
| Dating methods | *If new dates are provided, describe how they were obtained (e.g. collection, storage, sample pretreatment and measurement), where they were obtained (i.e. lab name), the calibration program and the protocol for quality assurance OR state that no new dates are provided.* |

☐ Tick this box to confirm that the raw and calibrated dates are available in the paper or in Supplementary Information.

| | |
|---|---|
| Ethics oversight | *Identify the organization(s) that approved or provided guidance on the study protocol, OR state that no ethical approval or guidance was required and explain why not.* |

Note that full information on the approval of the study protocol must also be provided in the manuscript.

# Animals and other research organisms

Policy information about studies involving animals; ARRIVE guidelines recommended for reporting animal research, and Sex and Gender in Research

| | |
|---|---|
| Laboratory animals | Wistar rats |
| Wild animals | N/A |
| Reporting on sex | N/A |
| Field-collected samples | N/A |
| Ethics oversight | All procedures with live animals were conducted in the animal facility at the Max-Planck-Institute for Multidisciplinary Sciences, Göttingen. According to the German Animal Welfare Law, killing is not an experiment on animals. All requirements of § 4 TierSchG together with § 2 Satz 2, Anlage 1, Abschnitt 2 and Anlage 2 TierSchVersV were implemented.<br><br>The facility is conducted under all aspects of animal welfare. The facility is headed by a veterinarian with special education in laboratory animal science as well as gene technology and molecular genetics.<br>Only professionally educated animal technicians are in charge of animal husbandry and care. The facility is registered according to §11 Abs. 1 TierSchG (Tierschutzgesetz der Bundesrepublik Deutschland, Animal Welfare Law of the Federal Republic of Germany) as documented by 33.23-42508-066-§11, dated Nov 16th, 2023 ("Erlaubnis, zum Halten von Wirbeltieren zur Versuchszwecken", "Permission to keep vertebrates for experimental purposes") by the Niedersächsisches Landesamt für Verbraucherschutz und Lebensmittelsicherheit (Lower Saxony State Office for Consumer Protection and Food Safety). According to the Animal Welfare Law of the Federal Republic of Germany (TierSchG) and the Regulation about animals used in experiments, dated 20th Dec 2022 (TierSchVersV) an animal welfare officer (specialized veterinarian in laboratory animal science) and an animal welfare committee for the institute is established. |

Note that full information on the approval of the study protocol must also be provided in the manuscript.

# Clinical data

Policy information about clinical studies

All manuscripts should comply with the ICMJE guidelines for publication of clinical research and a completed CONSORT checklist must be included with all submissions.

| | |
|---|---|
| Clinical trial registration | *Provide the trial registration number from ClinicalTrials.gov or an equivalent agency.* |
| Study protocol | *Note where the full trial protocol can be accessed OR if not available, explain why.* |
| Data collection | *Describe the settings and locales of data collection, noting the time periods of recruitment and data collection.* |
| Outcomes | *Describe how you pre-defined primary and secondary outcome measures and how you assessed these measures.* |

# Dual use research of concern

Policy information about dual use research of concern

## Hazards

Could the accidental, deliberate or reckless misuse of agents or technologies generated in the work, or the application of information presented in the manuscript, pose a threat to:

| No | Yes | |
|---|---|---|
| ☒ | ☐ | Public health |
| ☒ | ☐ | National security |
| ☒ | ☐ | Crops and/or livestock |
| ☒ | ☐ | Ecosystems |
| ☒ | ☐ | Any other significant area |

## Experiments of concern

Does the work involve any of these experiments of concern:

| No | Yes | |
|---|---|---|
| ⊠ | ☐ | Demonstrate how to render a vaccine ineffective |
| ⊠ | ☐ | Confer resistance to therapeutically useful antibiotics or antiviral agents |
| ⊠ | ☐ | Enhance the virulence of a pathogen or render a nonpathogen virulent |
| ⊠ | ☐ | Increase transmissibility of a pathogen |
| ⊠ | ☐ | Alter the host range of a pathogen |
| ⊠ | ☐ | Enable evasion of diagnostic/detection modalities |
| ⊠ | ☐ | Enable the weaponization of a biological agent or toxin |
| ⊠ | ☐ | Any other potentially harmful combination of experiments and agents |

# Plants

| | |
|---|---|
| Seed stocks | N/A |
| Novel plant genotypes | N/A |
| Authentication | N/A |

# ChIP-seq

## Data deposition

☐ Confirm that both raw and final processed data have been deposited in a public database such as GEO.

☐ Confirm that you have deposited or provided access to graph files (e.g. BED files) for the called peaks.

| | |
|---|---|
| Data access links<br>*May remain private before publication.* | *For "Initial submission" or "Revised version" documents, provide reviewer access links. For your "Final submission" document, provide a link to the deposited data.* |
| Files in database submission | *Provide a list of all files available in the database submission.* |
| Genome browser session<br>(e.g. UCSC) | *Provide a link to an anonymized genome browser session for "Initial submission" and "Revised version" documents only, to enable peer review. Write "no longer applicable" for "Final submission" documents.* |

## Methodology

| | |
|---|---|
| Replicates | *Describe the experimental replicates, specifying number, type and replicate agreement.* |
| Sequencing depth | *Describe the sequencing depth for each experiment, providing the total number of reads, uniquely mapped reads, length of reads and whether they were paired- or single-end.* |
| Antibodies | *Describe the antibodies used for the ChIP-seq experiments; as applicable, provide supplier name, catalog number, clone name, and lot number.* |
| Peak calling parameters | *Specify the command line program and parameters used for read mapping and peak calling, including the ChIP, control and index files used.* |
| Data quality | *Describe the methods used to ensure data quality in full detail, including how many peaks are at FDR 5% and above 5-fold enrichment.* |
| Software | *Describe the software used to collect and analyze the ChIP-seq data. For custom code that has been deposited into a community repository, provide accession details.* |

# Flow Cytometry

## Plots

Confirm that:

☒ The axis labels state the marker and fluorochrome used (e.g. CD4-FITC).

☒ The axis scales are clearly visible. Include numbers along axes only for bottom left plot of group (a 'group' is an analysis of identical markers).

☒ All plots are contour plots with outliers or pseudocolor plots.

☒ A numerical value for number of cells or percentage (with statistics) is provided.

## Methodology

| | |
|---|---|
| Sample preparation | U2OS cells were transfected with the bicistronic nuclease plasmids and the corresponding donor plasmids using FuGENE HD transfection reagent (Promega, Mannheim, Germany). To this end, 20.000 cells per well were seeded in a 6-well plate with supplemented DMEM. The following day, transfection was carried out using a reagent to DNA ratio of 3.5 to 1 and a total DNA amount of 3 µg. Subsequently, the cells were further incubated at 37°C, 5% CO2. After seven days, cells were inspected by fluorescence microscopy. Wells containing cells exhibiting the expected sub-cellular localization of the Dreiklang fusion protein were subjected to single cell sorting into 96-well plates. |
| Instrument | FACSAria II (BD Biosciences, Heidelberg, Germany) |
| Software | FACSDiva 6.0 software |
| Cell population abundance | 4.3% for PHB1-Dreiklang knockin cells (Fig. S2E, left); 10% for PHB2-Dreiklang knockin cells (Fig. S2E, right) |
| Gating strategy | We have identified Dreiklang-expressing knock-in cells by plotting the side scatter area versus fluorescence intensity after excitation with a 488 nm laser (Fig. S2D-E). We have sorted knock-in cells displaying a fluorescence signal that was about 5-fold higher compared to U2OS wildtype cells. |

☒ Tick this box to confirm that a figure exemplifying the gating strategy is provided in the Supplementary Information.

# Magnetic resonance imaging

## Experimental design

| | |
|---|---|
| Design type | *Indicate task or resting state; event-related or block design.* |
| Design specifications | *Specify the number of blocks, trials or experimental units per session and/or subject, and specify the length of each trial or block (if trials are blocked) and interval between trials.* |
| Behavioral performance measures | *State number and/or type of variables recorded (e.g. correct button press, response time) and what statistics were used to establish that the subjects were performing the task as expected (e.g. mean, range, and/or standard deviation across subjects).* |

## Acquisition

| | |
|---|---|
| Imaging type(s) | *Specify: functional, structural, diffusion, perfusion.* |
| Field strength | *Specify in Tesla* |
| Sequence & imaging parameters | *Specify the pulse sequence type (gradient echo, spin echo, etc.), imaging type (EPI, spiral, etc.), field of view, matrix size, slice thickness, orientation and TE/TR/flip angle.* |
| Area of acquisition | *State whether a whole brain scan was used OR define the area of acquisition, describing how the region was determined.* |

Diffusion MRI ☐ Used ☐ Not used

## Preprocessing

| | |
|---|---|
| Preprocessing software | *Provide detail on software version and revision number and on specific parameters (model/functions, brain extraction, segmentation, smoothing kernel size, etc.).* |
| Normalization | *If data were normalized/standardized, describe the approach(es): specify linear or non-linear and define image types used for transformation OR indicate that data were not normalized and explain rationale for lack of normalization.* |
| Normalization template | *Describe the template used for normalization/transformation, specifying subject space or group standardized space (e.g. original Talairach, MNI305, ICBM152) OR indicate that the data were not normalized.* |

| | |
|---|---|
| Noise and artifact removal | *Describe your procedure(s) for artifact and structured noise removal, specifying motion parameters, tissue signals and physiological signals (heart rate, respiration).* |
| Volume censoring | *Define your software and/or method and criteria for volume censoring, and state the extent of such censoring.* |

## Statistical modeling & inference

| | |
|---|---|
| Model type and settings | *Specify type (mass univariate, multivariate, RSA, predictive, etc.) and describe essential details of the model at the first and second levels (e.g. fixed, random or mixed effects; drift or auto-correlation).* |
| Effect(s) tested | *Define precise effect in terms of the task or stimulus conditions instead of psychological concepts and indicate whether ANOVA or factorial designs were used.* |

Specify type of analysis: ☐ Whole brain ☐ ROI-based ☐ Both

| | |
|---|---|
| Statistic type for inference<br><br>(See Eklund et al. 2016) | *Specify voxel-wise or cluster-wise and report all relevant parameters for cluster-wise methods.* |
| Correction | *Describe the type of correction and how it is obtained for multiple comparisons (e.g. FWE, FDR, permutation or Monte Carlo).* |

## Models & analysis

| n/a | Involved in the study |
|---|---|
| ☐ | ☐ Functional and/or effective connectivity |
| ☐ | ☐ Graph analysis |
| ☐ | ☐ Multivariate modeling or predictive analysis |

| | |
|---|---|
| Functional and/or effective connectivity | *Report the measures of dependence used and the model details (e.g. Pearson correlation, partial correlation, mutual information).* |
| Graph analysis | *Report the dependent variable and connectivity measure, specifying weighted graph or binarized graph, subject- or group-level, and the global and/or node summaries used (e.g. clustering coefficient, efficiency, etc.).* |
| Multivariate modeling and predictive analysis | *Specify independent variables, features extraction and dimension reduction, model, training and evaluation metrics.* |

