## [Peer Review File · Nature Cell Biology]

In-situ architecture of the human prohibitin complex

Corresponding Author: Professor Stefan Jakobs

Version 0:

Decision Letter:

*Please delete the link to your author homepage if you wish to forward this email to co-authors.

Dear Stefan,

Thank you again for submitting your manuscript, "In-situ architecture of the human prohibitin complex", to Nature Cell Biology and I am very sorry for the delay in sharing our decision with you, as one reviewer needed an extension. The manuscript has now been seen by 3 referees, who are experts in mitochondrial biology (Referee #1); cryoET (Referee #2); and mitochondria, quality control (Referee #3). As you will see from their comments (attached below), they found the work of potential interest but have raised substantial concerns, which in our view would need to be addressed with considerable revisions before we can consider publication in Nature Cell Biology.

As per our standard process, we have now discussed the referee reports in detail within the editorial team to identify key referee points that should be addressed with priority to strengthen the analyses and core conclusions. To guide the scope of the revisions, I have listed these points below. Our standard revision period is six months, and we are committed to providing a fair and constructive peer-review process, so please feel free to contact me if you would like to discuss any of the referee comments further or if you anticipate any issues addressing the reviews.

In particular, it would be essential to address the following points:

1- The reviewers had concerns related to the use of the fusion proteins and interpretation of the cryoET data that need to be addressed:

Rev#2 "- On lines 176-178, the authors should try to assign the MW to the DK densities. If you have 5-6, you would have 150KDa, does this approach the extra density they are seeing? Is it present in the terminus they tagged the protein with? From the nomenclature it should be the C-terminus, at the top of the bell shape, but the authors should specify. A schematic in the supplementary would be needed to rationalize this finding.

- While their KD data strongly suggests that the complex is not formed in the absence of one of the proteins, it is also common that knockdowns have other effects on protein complexes that do not contain the KD protein. The putative presence of DK densities is also somewhat satisfying, but a confidential discussion in my lab left some wanting clearer evidence. Some suggestions include:

- Would it be possible to predict a mutation at the interface between PHB1 and PHB2 and look for the complex not being present, or not pulled down?

- Is there an anti-DK nanobody that could be used to increase the size of the DK density? (I am not sure if the nano body would enter the mitochondria though). Alternative, can a larger tag be added? Can tags be added to both monomers?"

Rev#3 points #1-2-3-4

2- Please also address their questions about strengthening and clarifying existing analyses, minor points, technical comments, requests for text/figure edits or discussion.

3- Finally, please pay close attention to our guidelines on statistical and methodological reporting (listed below) as failure to do so may delay the reconsideration of the revised manuscript. In particular, please provide:

We would be happy to consider a revised manuscript that would satisfactorily address these points, unless a similar paper is published elsewhere or is accepted for publication in Nature Cell Biology in the meantime.

- ensure that it conforms to our format instructions and publication policies (see below and www.nature.com/nature/authors/).

- provide a point-by-point rebuttal to the full referee reports verbatim, as provided at the end of this letter.

- provide the completed Editorial Policy Checklist (found here <https://www.nature.com/authors/policies/Policy.pdf>), and Reporting Summary (found here <https://www.nature.com/authors/policies/ReportingSummary.pdf>). This is essential for reconsideration of the manuscript and these documents will be available to editors and referees in the event of peer review. For more information see <http://www.nature.com/authors/policies/availability.html> or contact me.

Nature Cell Biology is committed to improving transparency in authorship. As part of our efforts in this direction, we are now requesting that all authors identified as 'corresponding author' on published papers create and link their Open Researcher and Contributor Identifier (ORCID) with their account on the Manuscript Tracking System (MTS), prior to acceptance. ORCID helps the scientific community achieve unambiguous attribution of all scholarly contributions. You can create and link your ORCID from the home page of the MTS by clicking on 'Modify my Springer Nature account'. For more information please visit <http://www.springernature.com/orcid>.

Link Redacted

We would like to receive a revised submission within six months. We would be happy to consider a revision even after this timeframe, however if the resubmission deadline is missed and the paper is eventually published, the submission date will be the date when the revised manuscript was received.

We hope that you will find our referees' comments and editorial guidance helpful. Please do not hesitate to contact me if there is anything you would like to discuss. Thank you again for considering NCB for your work.

Best wishes,

Melina

Melina Casadio, PhD
Senior Editor, Nature Cell Biology
ORCID ID: <https://orcid.org/0000-0003-2389-2243>

Reviewers' Comments:

Reviewer #1:

Remarks to the Author:

This is an interesting, quantitative and rigorously performed study of mammalian prohibitin complexes in situ. Prior work indicated ~12-20 subunits of PHB1/PHB2 dimers form a ring in the inner mitochondrial membrane. The current cryo ET data define a discrete 11 subunit bell shaped complex resembling somewhat, to my eye, half of the major vault protein complex. Structural studies of a distantly related bacterial protein, HflK/C, that shares some domains with prohibitin, also reveals a similar bell shaped structure in a cell free reconstituted system. What this new work also brings is the structure of mammalian prohibitin in the native mitochondrial context. This new structural model differs substantially from the main current ring centric model of prohibitin oligomers. This new cage-like dome structure is also interesting because the center under the dome is empty, apparently not occupied by other proteins such as m-ATPases, and will likely inform/transform currently quite vague models for prohibitin function. The authors also find the prohibitin complex concentrated in nice quantitative analyses – in contrast to prior reports on the distribution of prohibitin in yeast. Overall, this is a highly rigorous advance into our understanding of prohibitin structure that will likely accelerate the understanding of function. The prior related structure of the bacterial protein HflK/C marginally decreases the novelty.

Specific comments:

On line 106 – State where the immunogold data is shown – Fig 1C-D.

In Fig 2D – please denote in panel 3 which face of the prohibitin bell faces the matrix and which side faces the intermembrane space.

Some of the valuable supplemental data could be incorporated into the main figures.

Reviewer #2:

Remarks to the Author:

The manuscript by Lange et al has a lot of qualities: the authors use state of the arts methods and quantitative calculations to design a project in which they can pursue the in situ structure of the elusive prohibitin complexes.

I really enjoyed following through the path of calculating the feasibility of the data acquisition by calculating mitochondrial surfaces, location of the complexes and endogenous copy numbers in cells, etc. I could only hope on a slam-dunk example like this when we wrote a recent Annual Reviews trying to make this very point.

The authors do everything I would have imagined that could be done for this project, and take advantage of AlphaFold to produce a model of the complex. The result is satisfying, and the first molecular model of prohibitin complexes. I think this is a fine example of the application of the cryo-tomography in situ (in cell) workflow, and it was fun to review. On the other hand, the project doesn't offer a mechanism or describes the context of prohibitins while performing their function. Nevertheless, I think this work is very well executed and I recommend it for publication in NCB.

I have a few minor and not-so minor comments that in my opinion will make the project more well rounded and will hopefully finish convincing a skeptical reviewer/reader.

Major:

- On lines 176-178, the authors should try to assign the MW to the DK densities. If you have 5-6, you would have 150KDa, does this approach the extra density they are seeing? Is it present in the terminus they tagged the protein with? From the nomenclature it should be the C-terminus, at the top of the bell shape, but the authors should specify. A schematic in the supplementary would be needed to rationalize this finding.

- While their KD data strongly suggests that the complex is not formed in the absence of one of the proteins, it is also common that knockdowns have other effects on protein complexes that do not contain the KD protein. The putative presence of DK densities is also somewhat satisfying, but a confidential discussion in my lab left some wanting clearer evidence. Some suggestions include:

- Would it be possible to predict a mutation at the interface between PHB1 and PHB2 and look for the complex not being present, or not pulled down?

- Is there an anti-DK nanobody that could be used to increase the size of the DK density? (I am not sure if the nano body would enter the mitochondria though). Alternative, can a larger tag be added? Can tags be added to both monomers?

- The confidence of the AlphaFold models should be included, and if not high in any area, the ability to use it should be justified. Molecular dynamics simulations from models typically have lower confidence than those from structural biology data.

- It was not clear to me how the authors decide on which of PHB1 or PHB2 should have 5 vs 6 monomers. Is this based on the copy numbers found in Figure 3? If so, it should be specified in the text.

- How do the authors reconcile the positive/negative alternating monomers with the C11 symmetry? How does the negative-negative or positive-positive interaction work? Why do the authors think about a mismatch from 11 to 12 compared to the bacterial system?

- It seems like the molecular dynamics model was membrane anchored at the bottom as it exists in the mitochondria, but this is only briefly described in the methods section but not on the figures. The termini needed to be contained in z, the authors should explain if the model is not enough to remain tethered to the membrane.

- Also, no information of on the reported stability of the molecular dynamics simulation is described. How was it assessed a feasible model?

Minor:

- Fluorescence images should be color-blind friendly

- For figure 1C, I would recommend that the squares around the gold labels are colored whether the gold is localized in the CM or IBM, preferably matching the colors of the bars in Fig. 1D.

- In line 122, is it 3.1×10^6 PHB1 and 9.8×10^6 PHB2 molecules per cell? The way it is written is confusing.

- Fig S7 - remove imod controls in A, caption should specify how particles in D were selected

- Fig 3. The numbers in D should have citations and original of computations in the caption.

Congratulations on the beautiful work, I hope our comments are helpful and constructive.

Elizabeth Villa and trainee

Reviewer #3:

Remarks to the Author:

The study by Lange, Ratz et al. reports on the first in situ cryo-EM structure of the human prohibitin complex. The authors use a combination of methods and an integrative structural approach to come up with a very interesting model on the human prohibitin complex. The data suggests that it forms a bell-shaped structure with around 20 nm diameter and 9 nm height which is consisting of 11 monomers of PHBs.

Overall this is a very interesting study and model of this complex which will be important in the mitochondrial field. It extends our knowledge on known 3D structures of this protein family considerably. I still feel that the refined 3D model after molecular modelling of 11 subunits of PHB into the bell-shaped complex is not substantiated sufficiently and the authors need to strengthen their model further.

Major issues:

1. Can the authors provide additional evidence of the proposed orientation and stoichiometry of 11 PHB subunits? Does their model fit to the density observed they claim to be caused by DK (Fig. S7D-E). What would MD simulations show in this case? The authors might consider to similar to their earlier approach also try to model the DK-containing fusion proteins into the observed densities.

2. Linked to this: Is it possible that these are actually 11 heterodimers fitting or can the authors exclude this with high certainty?

3. The functionality of the DK fusion constructs needs to be shown. The heterotypic interaction with endogenous PHBs is not sufficient in my opinion. A phenotypic rescue compared to the PHB knockout situation would be one possibility to show the functionality of the fusion protein.

Minor issues:

4. The authors have two arguments that the structure they observe is indeed PHB, namely the quantification of the PHB structures after downregulation and the additional density in the DK-containing complex (Fig. S7D-G). The latter is more convincing as the number of bell-shaped structures can result from indirect effects. Can the authors discuss whether the location of DK fits to the prediction where DK was integrated? Where is DK actually integrated exactly? Was this at the N- or C-terminus or internal? Please show a scheme where DK is relative to other PHB domains.
5. lines 98-101. The authors state that concentration of prohibitions is 3 to 5 times higher in the CM compared to the IBM. The calculation appears not to be correct when considering the ratio 0.64 to 1 (IBM to CM membrane length). According to my rough estimate the concentration (not amount) is rather 2 to 3.5 times higher.
6. lines 103-105. missing reference to Fig. 1CD. Please add.
7. Fig. 2D, 3AB, and Fig. S7F. These are very nice in situ images of the putative PHB complex. I suggest to include scale bars/size markers here to see the dimensions also in the figure.
8. Fig. S7C. The control with anti-PHB1 in Wt is missing. Please show
9. Fig S3D, right panel: mislabelling of x-axis as it should read "PHB2-DK". Please correct
10. Fig. S5. To ensure that this heterotypic interaction is indeed specific and not due to an unspecific "stickiness" of PHBs, it is needed to show that another inner membrane protein (e.g. CIV or CV subunit) is not interacting with PHB1-DK or PHB2-DK. It would further be good to show the fraction of unbound proteins. Please state how much was loaded of input and IP fractions.

Methods should be written concisely, but should contain all elements necessary to allow interpretation and replication of the results. As a

guideline, Methods sections typically do not exceed 3,000 words. The Methods should be divided into subsections listing reagents and techniques. When citing previous methods, accurate references should be provided and any alterations should be noted. Information must be provided about: antibody dilutions, company names, catalogue numbers and clone numbers for monoclonal antibodies; sequences of RNAi and cDNA probes/primers or company names and catalogue numbers if reagents are commercial; cell line names, sources and information on cell line identity and authentication. Animal studies and experiments involving human subjects must be reported in detail, identifying the committees approving the protocols. For studies involving human subjects/samples, a statement must be included confirming that informed consent was obtained. Statistical analyses and information on the reproducibility of experimental results should be provided in a section titled "Statistics and Reproducibility".

All Nature Cell Biology manuscripts submitted on or after March 21 2016 must include a Data availability statement at the end of the Methods section. For Springer Nature policies on data availability see <http://www.nature.com/authors/policies/availability.html>; for more information on this particular policy see <http://www.nature.com/authors/policies/data/data-availability-statements-data-citations.pdf>. The Data availability statement should include:

- Accession codes for primary datasets (generated during the study under consideration and designated as "primary accessions") and secondary datasets (published datasets reanalysed during the study under consideration, designated as "referenced accessions"). For primary accessions data should be made public to coincide with publication of the manuscript. A list of data types for which submission to community-endorsed public repositories is mandated (including sequence, structure, microarray, deep sequencing data) can be found here <http://www.nature.com/authors/policies/availability.html#data>.
- Unique identifiers (accession codes, DOIs or other unique persistent identifier) and hyperlinks for datasets deposited in an approved repository, but for which data deposition is not mandated (see here for details <http://www.nature.com/sdata/data-policies/repositories>).
- At a minimum, please include a statement confirming that all relevant data are available from the authors, and/or are included with the manuscript (e.g. as source data or supplementary information), listing which data are included (e.g. by figure panels and data types) and mentioning any restrictions on availability.
- If a dataset has a Digital Object Identifier (DOI) as its unique identifier, we strongly encourage including this in the Reference list and citing the dataset in the Methods.

We recommend that you upload the step-by-step protocols used in this manuscript to the Protocol Exchange. More details can found at www.nature.com/protocolexchange/about.

All imaging data should be accompanied by scale bars, which should be defined in the legend. Cropped images of gels/blots are acceptable, but need to be accompanied by size markers, and to retain visible background signal within the linear range (i.e. should not be saturated). The boundaries of panels with low background have to be demarked with black lines. Splicing of panels should only be considered if unavoidable, and must be clearly marked on the figure, and noted in the legend with a statement on whether the samples were obtained and processed simultaneously. Quantitative comparisons between samples on different gels/blots are discouraged; if this is unavoidable, it should only be performed for samples derived from the same experiment with gels/blots were processed in parallel, which needs to be stated in the legend.

Regardless of format, all figures must be vector graphic compatible files, not supplied in a flattened raster/bitmap graphics format, but

should be fully editable, allowing us to highlight/copy/paste all text and move individual parts of the figures (i.e. arrows, lines, x and y axes, graphs, tick marks, scale bars etc.). The only parts of the figure that should be in pixel raster/bitmap format are photographic images or 3D rendered graphics/complex technical illustrations.

The total number of Supplementary Figures (not including the "unprocessed scans" Supplementary Figure) should not exceed the number of main display items (figures and/or tables (see our Guide to Authors and March 2012 editorial <http://www.nature.com/ncb/authors/submit/index.html#suppinfo>; <http://www.nature.com/ncb/journal/v14/n3/index.html#ed>). No restrictions apply to Supplementary Tables or Videos, but we advise authors to be selective in including supplemental data.

GUIDELINES FOR EXPERIMENTAL AND STATISTICAL REPORTING

REPORTING REQUIREMENTS – To improve the quality of methods and statistics reporting in our papers we have recently revised the reporting checklist we introduced in 2013. We are now asking all life sciences authors to complete two items: an Editorial Policy Checklist (found here <https://www.nature.com/authors/policies/Policy.pdf>) that verifies compliance with all required editorial policies and a reporting summary (found here <https://www.nature.com/authors/policies/ReportingSummary.pdf>) that collects information on experimental design and reagents. These documents are available to referees to aid the evaluation of the manuscript. Please note that these forms are dynamic 'smart pdfs' and must therefore be downloaded and completed in Adobe Reader. We will then flatten them for ease of use by the reviewers. If you would like to reference the guidance text as you complete the template, please access these flattened versions at <http://www.nature.com/authors/policies/availability.html>.

STATISTICS – Wherever statistics have been derived the legend needs to provide the n number (i.e. the sample size used to derive statistics) as a precise value (not a range), and define what this value represents. Error bars need to be defined in the legends (e.g. SD, SEM) together with a measure of centre (e.g. mean, median). Box plots need to be defined in terms of minima, maxima, centre, and percentiles. Ranges are more appropriate than standard errors for small data sets. Wherever statistical significance has been derived, precise p values need to be provided and the statistical test used needs to be stated in the legend. Statistics such as error bars must not be derived from n<3. For sample sizes of n<5 please plot the individual data points rather than providing bar graphs. Deriving statistics from technical replicate samples, rather than biological replicates is strongly discouraged. Wherever statistical significance has been derived, precise p values need to be provided and the statistical test stated in the legend.

We strongly recommend the presentation of source data for graphical and statistical analyses as a separate Supplementary Table, and request that source data for all independent repeats are provided when representative experiments of multiple independent repeats, or averages of two independent experiments are presented. This supplementary table should be in Excel format, with data for different figures provided as different sheets within a single Excel file. It should be labelled and numbered as one of the supplementary tables,

titled "Statistics Source Data", and mentioned in all relevant figure legends.

Version 1:

Decision Letter:

Our ref: NCB-LE53553A

15th November 2024

Dear Dr. Jakobs,

Thank you for submitting your revised manuscript "In-situ architecture of the human prohibitin complex" (NCB-LE53553A) and for your patience with the process. It has now been seen by the original Referees #2-3 and their comments are below. The reviewers find that the paper has been strengthened in revision, and therefore we'll be happy in principle to publish it in Nature Cell Biology, pending minor revisions to comply with our editorial and formatting guidelines.

We are now performing detailed checks on your paper and will send you a checklist detailing our editorial and formatting requirements in about 1-2 weeks. Please do not upload the final materials and make any revisions until you receive this additional information from us.

Thank you again for your interest in Nature Cell Biology. Please do not hesitate to contact me if you have any questions.

Sincerely,

Melina

Melina Casadio, PhD
Senior Editor, Nature Cell Biology
Consulting Editor, Nature Structural & Molecular Biology
ORCID ID: <https://orcid.org/0000-0003-2389-2243>

Reviewer #2 (Remarks to the Author):

The authors address all the points raised by the reviewers. This is a great manuscript that we recommend for publication.

Reviewer #3 (Remarks to the Author):

The authors have well answered the open questions and addressed all concerns nicely. It is a nice and interesting study important to the field.

Version 2:

Decision Letter:

Dear Dr Jakobs,

I am pleased to inform you that your manuscript, "In-situ architecture of the human prohibitin complex", has now been accepted for publication in Nature Cell Biology.

Please note that *Nature Cell Biology* is a Transformative Journal (TJ). Authors may publish their research with us through the traditional subscription access route or make their paper immediately open access through payment of an article-processing charge (APC). Authors will not be required to make a final decision about access to their article until it has been accepted. [Find out more about Transformative Journals](https://www.springernature.com/gp/open-research/transformative-journals)

If you have not already done so, we strongly recommend that you upload the step-by-step protocols used in this manuscript to protocols.io (<https://protocols.io>), an open online resource that allows researchers to share their detailed experimental know-how. All uploaded protocols are made freely available and are assigned DOIs for ease of citation. Protocols and Nature Portfolio journal papers in which they are used can be linked to one another, and this link is clearly and prominently visible in the online versions of both. Authors who performed the specific experiments can act as primary authors for the Protocol as they will be best placed to share the methodology details, but the Corresponding Author of the present research paper should be included as one of the authors. By uploading your Protocols onto protocols.io, you are enabling researchers to more readily reproduce or adapt the methodology you use, as well as increasing the visibility of your protocols and papers. You can also establish a dedicated workspace to collect your lab Protocols. Further information can be found at <https://www.protocols.io/help/publish-articles>.

Nature Cell Biology encourages authors presenting evidence for cell, biological, molecular, and genetic interactions to consider communicating these findings using Biofactoid (<https://biofactoid.org/>). This tool helps users share a searchable representation of interactions (e.g. binding, gene expression, post-translational modification) between genes, gene products, or chemicals. Information added to Biofactoid, with author attribution, is shared on social media and public databases, such as Pathway Commons, where it can be discovered and analyzed in the context of a large and growing corpus of knowledge.

With kind regards,

Melina Casadio, PhD
Senior Editor, Nature Cell Biology
Consulting Editor, Nature Structural & Molecular Biology
ORCID ID: <https://orcid.org/0000-0003-2389-2243>

** Visit the Springer Nature Editorial and Publishing website at <http://editorial-jobs.springernature.com?>

utm_source=ejp_NCB_email&utm_medium=ejp_NCB_email&utm_campaign=ejp_NCB">www.springernature.com/editorial-and-publishing-jobs for more information about our career opportunities. If you have any questions please click here.**

Dear Reviewers, dear Dr. Casadio,

Thank you for your constructive comments and the positive views on our manuscript. We have taken the reviewers' comments very seriously and believe that with the revised version of our manuscript we are able to address the raised concerns appropriately.

We added an entirely new dataset of crosslinking mass spectrometry analysis and a new set of cryo electron tomography data. Thereby, we added two entirely new figures (Main Fig. 4 and Suppl. Fig. 10), re-wrote the manuscript and made extensive adjustments and additions to existing figures.

We are convinced that the revised version of our manuscript is now suitable for publication in NCB.

Please find below a detailed point-by-point response to the referee reports.

Yours sincerely,

Reviewers' Comments:

Reviewer #1:

Remarks to the Author:

This is an interesting, quantitative and rigorously performed study of mammalian prohibitin complexes in situ. Prior work indicated ~12-20 subunits of PHB1/PHB2 dimers form a ring in the inner mitochondrial membrane. The current cryo ET data define a discrete 11 subunit bell shaped complex resembling somewhat, to my eye, half of the major vault protein complex. Structural studies of a distantly related bacterial protein, HflK/C, that shares some domains with prohibitin, also reveals a similar bell shaped structure in a cell free reconstituted system. What this new work also brings is the structure of mammalian prohibitin in the native mitochondrial context. This new structural model differs substantially from the main current ring centric model of prohibitin oligomers. This new cage-like dome structure is also interesting because the center under the dome is empty, apparently not occupied by other proteins such as m-ATPases, and will likely inform/transform currently quite vague models for prohibitin function. The authors also find the prohibitin complex concentrated in cristae in nice quantitative analyses – in contrast to prior reports on the distribution of prohibitin in yeast. Overall, this is a highly rigorous advance into our understanding of prohibitin structure that will likely accelerate the understanding of function. The prior related structure of the bacterial protein HflK/C marginally decreases the novelty.

We thank reviewer #1 for his/her positive view on our manuscript and the constructive comments and suggestions.

Specific comments:

On line 106 –State where the immunogold data is shown – Fig 1C-D.

Thank you for pointing to this. We corrected the text accordingly.

In Fig 2D – please denote in panel 3 which face of the prohibitin bell faces the matrix and which side faces the intermembrane space.

We thank the reviewer for this suggestion. We modified the respective figure panel (Fig. 2D3) and denoted the respective mitochondrial compartments as *mitochondrial matrix* and *IMS/crista lumen*, respectively.

Some of the valuable supplemental data could be incorporated into the main figures.

This is an interesting suggestion. We feel that the manuscript is concise in its current form and would prefer to keep it that way. It is also within the length guidelines of NatCellBiol. We would prefer to leave it to the discretion of the journal editors to decide whether further supplementary data should be included in the main figures.

Reviewer #2:

Remarks to the Author:

The manuscript by Lange et al has a lot of qualities: the authors use state of the arts methods and quantitative calculations to design a project in which they can pursue the in situ structure of the elusive prohibitin complexes.

I really enjoyed following through the path of calculating the feasibility of the data acquisition by calculating mitochondrial surfaces, location of the complexes and endogenous copy numbers in cells, etc. I could only hope on a slam-dunk example like this when we wrote a recent Annual Reviews trying to make this very point.

The authors do everything I would have imagined that could be done for this project, and take advantage of AlphaFold to produce a model of the complex. The result is satisfying, and the first molecular model of prohibitin complexes. I think this is a fine example of the application of the cryo-tomography in situ (in cell) workflow, and it was fun to review. On the other hand, the project doesn't offer a mechanism or describes the context of prohibitins while performing their function. Nevertheless, I think this work is very well executed and I recommend it for publication in NCB.

We thank reviewer #2 for her positive view on our manuscript and the constructive comments and suggestions.

I have a few minor and not-so minor comments that in my opinion will make the project more well rounded and will hopefully finish convincing a skeptical reviewer/reader.

Major:

- On lines 176-178, the authors should try to assign the MW to the DK densities. If you have 5-6, you would have 150KDa, does this approach the extra density they are seeing? Is it present in the terminus they tagged the protein with? From the nomenclature it should be the C-terminus, at the top of the bell shape, but the authors should specify. A schematic in the supplementary would be needed to rationalize this finding.

As suggested by the reviewer, we added a schematic in the supplementary to rationalize the orientation of the prohibitins in the inner membrane (Fig. S7E). This figure highlights that DK fused to the C-terminus of PHB1 or PHB 2 is located in the inter membrane space.

The question on how many DKs we do see in the densities is a difficult one: Because both tagged cell lines (PHB1-DK and PHB2-DK) were heterozygous, we expect various numbers of DK molecules in the single complexes. Indeed, in the tomograms we could distinguish 1, 2 and 3 DK molecules and the maximum we were able to see was 5 molecules on top of a complex.

This is now explained in the revised version of the manuscript. It reads:

“We also observed one to three additional densities at most (> 90%) convex structures identified in the tomograms of U2OS PHB1-DK and PHB2-DK cells (Fig. S7E-H). These extra densities sat at the top of the convex structure but were absent in wildtype cells, thereby indicating that they originate from the DK fluorescent protein fused to the C-termini of the PHB1 and PHB2 proteins.” (page 5)

- While their KD data strongly suggests that the complex is not formed in the absence of one of the proteins, it is also common that knockdowns have other effects on protein complexes that do not contain the KD protein. The putative presence of DK densities is also somewhat satisfying, but a confidential discussion in my lab left some wanting clearer evidence. Some suggestions include:

- Would it be possible to predict a mutation at the interface between PHB1 and PHB2 and look for the complex not being present, or not pulled down?

We thank the reviewer for this interesting idea. To the best of our knowledge no mutations to that effect have been described. We are not able to make a prediction for such a mutation based on the structure. For these reasons we could not follow this suggestion.

- Is there an anti-DK nanobody that could be used to increase the size of the DK density? (I am not sure if the nano body would enter the mitochondria though).

We thank the reviewer for this interesting idea. We thought about it carefully. Several studies address the live cell delivery of binding molecules (e.g. antibodies) as demonstrated in Kai W. Teng et al.; eLife 2016 or pre-embedding labelling of cytoplasmic targets as demonstrated in Elena V. Polishchuk et al.; Tissue and Cell 2019. While the former would technically be compatible with vitrification and near-native structural preservation the latter is likely not. These techniques have in common that they work on cytoplasmic proteins and studies showing this principle within mitochondria are lacking. To the best of our knowledge there are currently no tools available to facilitate temporary permeabilization or other micropores of the mitochondrial membranes in live cells. We therefore are not able to pursue this idea.

- Alternative, can a larger tag be added?

We thank the reviewer also for this interesting suggestion. Indeed, the identification of novel protein complexes poses a challenge for *in-situ* structural biology studies. Recent developments to identify proteins of interest in cryo electron tomograms include the GEM tag from the Mahamid lab (Herman K. H. Fung et al. 2023; Nature Methods) and the FerriTag from the Royle lab (Nicholas I. Clarke et al. 2018; Nature Communications). As of now, none of these tags were shown to work in mitochondria. Further, both tags require a substantial number of monomers (60 monomers for GEM, 24 monomers for ferritin). Hence, it remains unclear whether such a large particle (>15 nm in diameter) could be assembled within the narrow space of the crista lumen or the intermembrane space of mitochondria. Hence, we doubt that this approach would work with the prohibitin complex. As described in the manuscript, we decided instead to tag also PHB1 in order to provide additional proof that the observed complexes are indeed prohibitin complexes.

- Can tags be added to both monomers?

For the revised version of the manuscript, we tagged both prohibitins (PHB1 and PHB2) individually with DK. For the revision, we also imaged PHB1-DK and found extra densities on top of the prohibitin complexes. Comparable to the complexes in PHB2-DK cells, we recorded populations of the complexes showing 1, 2 or 3 densities that represent individual DK molecules (of about 3 nm size). This data is now included in Figure S7F.

- The confidence of the AlphaFold models should be included, and if not high in any area, the ability to use it should be justified. Molecular dynamics simulations from models typically have lower confidence than those from structural biology data.

We thank the reviewer for this comment. We added additional panels to Figure S9A to highlight the AlphaFold-predicted structures of PHB1 and PHB2 colored according to the pLDDT. Due to the structural similarities among SPFH protein family members and the availability of in-vitro structures for the bacterial homolog HflK/C, the pLDDT scores are high for the majority of the two prohibitins. . pLDDT values < 50 are only present in the flexible part of the C-terminus. To our understanding, the current state of subtomogram averaging approaches is insufficient in capturing small, unstructured and potentially very flexible peptides of a protein and therefore we did not draw any conclusions for this part of the PHB complex.

- It was not clear to me how the authors decide on which of PHB1 or PHB2 should have 5 vs 6 monomers. Is this based on the copy numbers found in Figure 3? If so, it should be specified in the text.

We believe that both options (5 PHB1 vs 6 PHB2, or 6 PHB1 vs 5 PHB2) exist. This is now clearly stated in the manuscript and shown in the new figures Figure 3A and Figure S10B. This is also stated in the text. It reads:

“The final molecular model comprises 11 monomeric PHB molecules forming a bell-shaped assembly (Fig. 3A1, 2, Fig. S9B, C). Obviously, the uneven symmetry implies that one pair of adjacent subunits comprises a repeat of the same molecule (PHB1-PHB1 or PHB2-PHB2).” (page 7)

- How do the authors reconcile the positive/negative alternating monomers with the C11 symmetry? How does the negative-negative or positive-positive interaction work? Why do the authors think about a mismatch from 11 to 12 compared to the bacterial system?

We thank the reviewer for this comment.

To address the question on how the negative-negative or positive-positive interactions work, we analysed for the revised version crosslinking mass spectrometry data (new Fig. 4 and Fig. S10A). The data demonstrate that PHB1-PHB1 as well as PHB2-PHB2 interactions exist, although PHB1-PHB2 interactions are much more abundant, fully supporting the proposed model of a prohibitin structure consisting of an 11-mer of alternating PHB1 and PHB2 molecules.

The C11 symmetry comes out of the data and has not been imposed by us.

This is stated in the revised manuscript:

“We proceeded to extract the CTF corrected subtomograms in Warp for subsequent automated 3D refinement in Relion4.0 with no initial symmetry implied^{25, 26}. Initially, we aligned all particles at 5 Å pixel size. The resulting cryo EM map revealed 11 densities in the top view likely representing subunits of the prohibitin complex (Fig. S8C).” (page 6)

And

“Pose-optimized particles were extracted in Warp at a 5 Å pixel size and aligned in Relion4.0 through 3D auto-refinement, revealing an 11-fold symmetry of the prohibitin complex.” (page 30)

In fact, we do not consider the stoichiometry of the complex to be a mismatch to the bacterial homologue: While the monomers of the SPFH members share substantial structural similarities their multimeric complexes show strikingly different dimensions. The RNA-vault for example comprises of 39 molecules, the bacterial HflK/C contains 24 molecules and two recent studies addressing the structure of flotillin in-vitro and in-situ suggested a range of 42 to 44 molecules per complex. All of these examples result in protein complexes that are substantially larger compared to the PHB complex.

Therefore, the stoichiometry and dimension of the complex is possibly be dictated by the specific function as well as the available space, which is limited in the case of prohibitins compared to cytoplasmic complexes such as the RNA-vault and flotillin. Because this is rather speculative, we prefer not to address this issue in the manuscript.

- It seems like the molecular dynamics model was membrane anchored at the bottom as it exists in the mitochondria, but this is only briefly described in the methods section but not on the figures. The termini needed to be contained in z, the authors should explain if the model is not enough to remain tethered to the membrane.

We thank the reviewer for this suggestion. The molecular dynamics simulation did not include any restraints or anchoring. The restraints on the N-termini position in the Z-axis were only applied during the initial modelling phase, prior to embedding the protein complex in the membrane and preparing it for molecular dynamics. These initial restraints were used to ensure that the newly modelled N-termini were not positioned in the region where the lipid bilayer would later be constructed. Indeed, the model by itself remained in the lipid bilayer during the MD simulation without any enforcements. These points are now explained in the manuscript:

“Ultimately, because biochemical evidence suggests a complex assembly with alternating PHB1 and PHB2 molecules³, successive PHB1 and PHB2 molecules were fitted into the cryo-EM map with the N-terminal transmembrane domains embedded in the lipid bilayer³ and the PHB domains perpendicular to the lipid bilayer.” (page 7)

- Also, no information of on the reported stability of the molecular dynamics simulation is described. How was it assessed a feasible model?

The reviewer is right. In the initial submission we only touched on the stability briefly. We added the missing information to the manuscript in Figure S9F and modified the text accordingly. It now reads:

“The molecular model remained stable over 100 ns of MD simulation without major alterations, suggesting that it represents a plausible model for the molecular structure of the prohibitin complex in human cells.” (page 8)

Minor:

- Fluorescence images should be color-blind friendly

The reviewer is right. To aid readability of the manuscript and figures we changed the lookup tables of the dual-color fluorescence microscopy images to green/magenta.

- For figure 1C, I would recommend that the squares around the gold labels are colored whether the gold is localized in the CM or IBM, preferably matching the colors of the bars in Fig. 1D.

Done. We applied this change and the boxes in Figure 1 C now match the color of the bars shown in Figure 1D.

- In line 122, is it 3.1×10^6 PHB1 and 9.8×10^6 PHB2 molecules per cell ? The way it is written is confusing.

The reviewer is right. We clarified this statement. It now reads:

“To this end, recombinant His-tagged PHB1 and PHB2 proteins were purified from Escherichia coli and used as a reference in quantitative Western blotting (Fig. S6B). It was determined that on average each U2OS cell contains approximately $3.38 \pm 0.23 \cdot 10^6$ molecules of PHB1 and $3.46 \pm 0.15 \cdot 10^6$ molecules of PHB2. These numbers are in good agreement with previous mass spectrometry studies using various human cell lines^{17, 18}.” (page 4)

- Fig S7 - remove imod controls in A, caption should specify how particles in D were selected
Done. In the revised manuscript we removed the IMOD controls in Fig. S7A and indicated the change of perspective with an arrow in Panel A2. Further, the sub-title of Fig. S7D now reads "*Manual annotation of prohibitin complexes*".
- Fig 3. The numbers in D should have citations and original of computations in the caption.
Done. In the revised manuscript we modified the caption of Fig. 5 which now reads "**B:** *Mitochondrial key numbers and prohibitin abundance determined in this study. Mitochondrial network length determined in Fig. S6C, inter-cristae distance determined in Fig. S6D-F, PHB molecules per cell determined in Fig. S6B.*"

Congratulations on the beautiful work, I hope our comments are helpful and constructive.

Elizabeth Villa and trainee

Reviewer #3:

Remarks to the Author:

The study by Lange, Ratz et al. reports on the first in situ cryo-EM structure of the human prohibitin complex. The authors use a combination of methods and an integrative structural approach to come up with a very interesting model on the human prohibitin complex. The data suggests that it forms a bell-shaped structure with around 20 nm diameter and 9 nm height which is consisting of 11 monomers of PHBs.

Overall this is a very interesting study and model of this complex which will be important in the mitochondrial field. It extends our knowledge on known 3D structures of this protein family considerably. I still feel that the refined 3D model after molecular modelling of 11 subunits of PHB into the bell-shaped complex is not substantiated sufficiently and the authors need to strengthen their model further.

We thank reviewer #3 for his/her positive view on our manuscript and the constructive comments and suggestions.

Major issues:

1. Can the authors provide additional evidence of the proposed orientation and stoichiometry of 11 PHB subunits? Does their model fit to the density observed they claim to be caused by DK (Fig. S7D-E). What would MD simulations show in this case? The authors might consider to similar to their earlier approach also try to model the DK-containing fusion proteins into the observed densities.

The C11 symmetry comes out of the data and has not been imposed by us. We just do not see any possible way to provide additional evidence for a C11 symmetry.

In order to provide additional evidence for the proposed stoichiometry of the PHB subunits we analysed for the revised version crosslinking mass spectrometry data (new Fig. 4 and Fig. S10A). The data demonstrate that PHB1-PHB1 as well as PHB2-PHB2 interactions exist, although PHB1-PHB2 interactions are much more abundant, fully supporting the proposed model of a prohibitin structure consisting of an 11-mer of alternating PHB1 and PHB2 molecules.

Concerning the proposed orientation in the mitochondrial inner membrane: Our initial model design was based on the bacterial homologue HflK/C, which demonstrated N-terminal anchoring of the individual molecules in the lipid bilayer. Indeed, the predicted transmembrane domain (TMD) of prohibitins is also at the N-terminus. The final cryoEM-map of the prohibitin structure reveals a large domain close to the lipid bilayer which indicates the location of the PHB-domain. Finally, our tomography data of PHB1-DK and PHB2-DK cells, both tagged at the C-terminus demonstrated the additional densities at the top of the convex structures indicating that the prohibitin C-termini form the top of the structures. To clarify the orientation, we added an additional panel to Fig. S7 (Fig. S7E) to illustrate the location of DK fused to PHB2.

Both cell lines (PHB1-DK and PHB2-DK) were heterozygous. Therefore, we do expect various numbers of DK molecules being present on top of the complexes. This is exactly what the tomograms show. We identified typically one to three extra densities on top of the prohibitin complexes in the PHB1-DK and PHB2-DK heterozygote cell lines. Due to the flexibility of the C-terminal tag (linker and DK), a sharp density is not expected (and not observed) upon subtomogram averaging. Consequently, the density achieved in the simple subtomogram averaging approach presented in Fig. S7G is only slightly larger than a single DK-molecule. To present this finding clearer we

changed the simple subtomogram average to a transparent surface and fitted the structure of DK into the density (New panel in Fig. S7G).

Because of the variable number of DK-molecules and their flexibility it was possible to achieve a sharpened, higher-resolution cryo-EM map of the DK-PHB complexes. Consequently, we did not consider MD simulations.

This is also better explained in the text of the revised version.

It now reads: *“We also observed one to three additional densities in most (> 90%) convex structures identified in the tomograms of U2OS PHB1-DK and PHB2-DK cells (Fig. S7E-H). These extra densities sat at the top of the convex structure but were absent in wildtype cells, thereby indicating that they originate from the DK fluorescent protein fused to the C-termini of the PHB1 and PHB2 proteins.”* (page 5)

2. Linked to this: Is it possible that these are actually 11 heterodimers fitting or can the authors exclude this with high certainty?

We can exclude with high certainty the existence of 11 heterodimers: Heterodimers just do not fit into the high-resolution EM-Map. To explain the conclusion on the C11 symmetry, we also added new panels to Fig. S9 that highlight the rationale of the stoichiometry and the initial molecule placement in more detail (Fig. S9C).

This is now clearly stated in the revised version:

“Numerous possible arrangements were tested but we found that each of the 11 densities in the cryo-EM map can accommodate only one prohibitin molecule. Therefore, the entire complex consists of 11 prohibitin molecules in total.” (page 6)

3. The functionality of the DK fusion constructs needs to shown. The heterotypic interaction with endogenous PHBs is not sufficient in my opinion. A phenotypic rescue compared to the PHB knockout situation would be one possibility to show the functionality of the fusion protein.

Here we respectfully disagree with the reviewer. In this study, we did not draw any functional conclusions from the knock-in cell lines other than the mobility of the complexes (see also Figure 1E and Figure 1F).

We use the tagged versions only as part of a line of evidence to demonstrate that the bell-shaped structure is indeed the prohibition complex. We believe that our line of evidence is exhaustive.

We therefore argue that a more detailed functional analysis is not necessary at this stage of the work.

Minor issues:

4. The authors have two arguments that the structure they observe is indeed PHB, namely the quantification of the PHB structures after downregulation and the additional density in the DK-containing complex (Fig. S7D-G). The latter is more convincing as the number of bell-shaped structures can result from indirect effects. Can the authors discuss whether the location of DK fits to the prediction where DK was integrated? Where is DK actually integrated exactly? Was this at the N- or C-terminus or internal? Please show a scheme where DK is relative to other PHB domains.

We thank the reviewer for this suggestion. We agree with the reviewer that both experiments, PHB knock-down and cryo-ET imaging of PHB-DK complexes, are necessary. DK was added after a short GSGSG linker to the C-terminus of PHB1 and PHB2. Therefore, the location of the extra densities in cryo-ET corresponds exactly to the expected position, taking into account the N-terminal integration of the prohibitin molecules into the lipid bilayer. In the revised manuscript, we have added the panel Fig. S7E to illustrate this rationale more clearly.

5. lines 98-101. The authors state that concentration of prohibitions is 3 to 5 times higher in

the CM compared to the IBM. The calculation appears not to be correct when considering the ratio 0.64 to 1 (IBM to CM membrane length). According to my rough estimate the concentration (not amount) is rather 2 to 3.5 times higher.

We thank the reviewer for this comment. The analysis of immunogold EM data presented in Fig. 1D was already adjusted for the IBM to CM ratio. Therefore, and to make this section clearer we rephrased the section which now reads:

“For PHB1-DK and PHB2-DK we found 84.9% and 90.6% of the gold particles at the crista membranes, respectively; with antibodies directed against endogenous PHB1 and PHB2 we found 88.6% and 88.9% of the gold particles at the crista membranes, respectively (Fig. 1C). When considering that the ratio of IBM to CM is 0.64 to 1 in the U2OS cells (Fig. S6A), it can be calculated that the concentration of prohibitins is three to five times higher in the crista membranes than in the IBM (Fig. 1D).” (page 3)

6. lines 103-105. missing reference to Fig. 1CD. Please add.

Thank you for pointing to this. We corrected the text accordingly.

7. Fig. 2D, 3AB, and Fig. S7F. These are very nice in situ images of the putative PHB complex. I suggest to include scale bars/size markers here to see the dimensions also in the figure.

We thank the reviewer for this suggestion. We added scale bars to Fig. 2D1,2 and Fig. S7G to highlight the dimensionality of the complexes better.

8. Fig. S7C. The control with anti-PHB1 in Wt is missing. Please show

The reviewer is right. We added the missing control which is now incorporated into Fig. S7C.

9. Fig S3D, right panel: mislabelling of x-axis as it should read “PHB2-DK”. Please correct

The reviewer is right. We changed the labelling of the x-axis which now reads “*PHB2-DK clone #*”

10. Fig. S5. To ensure that this heterotypic interaction is indeed specific and not due to an unspecific “stickiness” of PHBs, it is needed to show that another inner membrane protein (e.g. CIV or CV subunit) is not interacting with PHB1-DK or PHB2-DK. It would further be good to show the fraction of unbound proteins. Please state how much was loaded of input and IP fractions.

The reviewer is right and we thank for this suggestion. In the revised version we shown two other inner membrane proteins (ATP5A und COX2) and also show the fraction of unbound proteins (Fig. S5B). We also state how much was loaded of input and IP fractions.

In the figure legend to Fig. S5 it reads: “*10 µg of sample was loaded for all samples of Input and flow through, respectively.*”

From the editor:

1- The reviewers had concerns related to the use of the fusion proteins and interpretation of the cryoET data that need to be addressed:

Rev#2

- On lines 176-178, the authors should try to assign the MW to the DK densities. If you have 5-6, you would have 150KDa, does this approach the extra density they are seeing? Is it present in the terminus they tagged the protein with? From the nomenclature it should be the C-terminus, at the top of the bell shape, but the authors should specify. A schematic in the supplementary would be needed to rationalize this finding.

As suggested by the reviewer, we added a schematic in the supplementary to rationalize the orientation of the prohibitins in the inner membrane (Fig. S7E).

In the revised version of the manuscript, we further modified line 174 to 178. It reads: *"We also observed one to three additional densities at most (> 90%) convex structures identified in the tomograms of U2OS PHB1-DK and PHB2-DK cells (Fig. S7E-H). These extra densities sat at the top of the convex structure but were absent in wildtype cells, thereby indicating that they originate from the DK fluorescent protein fused to the C-termini of the PHB1 and PHB2 proteins."*

- While their KD data strongly suggests that the complex is not formed in the absence of one of the proteins, it is also common that knockdowns have other effects on protein complexes that do not contain the KD protein. The putative presence of DK densities is also somewhat satisfying, but a confidential discussion in my lab left some wanting clearer evidence. Some suggestions include:

- Would it be possible to predict a mutation at the interface between PHB1 and PHB2 and look for the complex not being present, or not pulled down?

To the best of our knowledge no mutations to that effect have been described. We are not able to make a prediction for such a mutation based on the structure. For these reasons we could not follow this suggestion.

- Is there an anti-DK nanobody that could be used to increase the size of the DK density? (I am not sure if the nano body would enter the mitochondria though). Alternative, can a larger tag be added? Can tags be added to both monomers?

We thought about it carefully but to the best of our knowledge there are currently no tools available to facilitate temporary permeabilization or other micropores of the mitochondrial membranes in live cells. We therefore are not able to pursue this idea.

Rev#3 points #1-2-3-4

1. Can the authors provide additional evidence of the proposed orientation and stoichiometry of 11 PHB subunits? Does their model fit to the density observed they claim to be caused by DK (Fig. S7D-E). What would MD simulations show in this case? The authors might consider to similar to their earlier approach also try to model the DK-containing fusion proteins into the observed densities.

In order to provide additional evidence for the proposed stoichiometry of the PHB subunits, we analysed for the revised version crosslinking mass spectrometry data (new Fig. 4 and new Fig. S10A). The data demonstrate that PHB1-PHB1 as well as PHB2-PHB2 interactions exist, although PHB1-PHB2 interactions are much more abundant, fully supporting the proposed model of a prohibitin structure consisting of an 11-mer of alternating PHB1 and PHB2 molecules.

Concerning the proposed orientation in the mitochondrial inner membrane: To clarify the orientation, we added an additional panel to Fig. S7 to illustrate the location of DK fused to PHB2 (Fig S7E).

Because of the variable number of DK-molecules it is not possible to achieve a sharpened, higher-resolution cryo-EM map of the DK-PHB complexes. Consequently, we did not consider MD simulations. This is also better explained in the text of the revised version. It now reads:

“We also observed one to three additional densities at most (> 90%) convex structures identified in the tomograms of U2OS PHB1-DK and PHB2-DK cells (Fig. S7E-H). These extra densities sat at the top of the convex structure but were absent in wildtype cells, thereby indicating that they originate from the DK fluorescent protein fused to the C-termini of the PHB1 and PHB2 proteins.” (Line 174 to 178)

To present this finding clearer we changed the simple subtomogram average to a transparent surface and fitted the structure of DK into the density (New panel in Fig. S7G).

2. Linked to this: Is it possible that these are actually 11 heterodimers fitting or can the authors exclude this with high certainty?

We can exclude with high certainty the existence of 11 heterodimers: Heterodimers just do not fit into the high-resolution EM-Map. To explain the conclusion on the C11 symmetry, we also added new panels to Fig. S9 that highlight the rational of the stoichiometry and the initial molecule placement in more detail (Fig. S9C).

This is now clearly stated in the revised version:

“Numerous possible arrangements were tested but we found that each of the 11 densities in the cryo-EM map can accommodate only one prohibitin molecule. Therefore, the entire complex consists of 11 prohibitin molecules in total.” (Line 215 to 217)

3. The functionality of the DK fusion constructs needs to shown. The heterotypic interaction with endogenous PHBs is not sufficient in my opinion. A phenotypic rescue compared to the PHB knockout situation would be one possibility to show the functionality of the fusion protein.

In this study, we did not draw any functional conclusions from the knock-in cell lines other than the mobility of the complexes.

We use the tagged versions only as part of a line of evidence to demonstrate that the bell-shaped structure is indeed the prohibitin complex. We believe that our line of evidence is exhaustive. We therefore argue that a more detailed functional analysis is not necessary at this stage of the work.

4. The authors have two arguments that the structure they observe is indeed PHB, namely the quantification of the PHB structures after downregulation and the additional density in the DK-containing complex (Fig. S7D-G). The latter is more convincing as the number of bell-shaped structures can result from indirect effects. Can the authors discuss whether the location of DK fits to the prediction where DK was integrated? Where is DK actually integrated exactly? Was this at the N- or C-terminus or internal? Please show a scheme where DK is relative to other PHB domains.

We agree with the reviewer that both experiments, PHB knock-down and cryo-ET imaging of PHB-DK complexes, are necessary. DK was added after a short linker to the C-terminus of PHB1 and PHB2. Therefore, the location of the extra densities in cryo-ET corresponds exactly to the expected position, taking into account the N-terminal integration of the prohibitin molecules into the lipid bilayer. In the revised manuscript, we have added the panel Fig. S7E to illustrate this rationale more clearly.

2- Please also address their questions about strengthening and clarifying existing analyses, minor points, technical comments, requests for text/figure edits or discussion.

We are convinced that we addressed all questions appropriately.

3- Finally, please pay close attention to our guidelines on statistical and methodological reporting (listed below) as failure to do so may delay the reconsideration of the revised manuscript. In particular, please provide:

Done.

Done.